EMBO
reports

# Dynamics of *Staphylococcus aureus* Cas9 in DNA target Association and Dissociation

Siqi Zhang[1,2,3,†], Qian Zhang[1,2,3,†] (ID), Xi-Miao Hou[4], Lijuan Guo[1,2,3], Fangzhu Wang[5], Lulu Bi[1] (ID), Xia Zhang[1], Hai-Hong Li[4], Fengcai Wen[1,2,3], Xu-Guang Xi[6], Xingxu Huang[1], Bin Shen[5] & Bo Sun[1,*] (ID)

## Abstract

*Staphylococcus aureus* Cas9 (SaCas9) is an RNA-guided endonuclease that targets complementary DNA adjacent to a protospacer adjacent motif (PAM) for cleavage. Its small size facilitates in vivo delivery for genome editing in various organisms. Herein, using single-molecule and ensemble approaches, we systemically study the mechanism of SaCas9 underlying its interplay with DNA. We find that the DNA binding and cleavage of SaCas9 require complementarities of 6- and 18-bp of PAM-proximal DNA with guide RNA, respectively. These activities are mediated by two steady interactions among the ternary complex, one of which is located approximately 6 bp from the PAM and beyond the apparent footprint of SaCas9 on DNA. Notably, the other interaction within the protospacer is significantly strong and thus poses DNA-bound SaCas9 a persistent block to DNA-tracking motors. Intriguingly, after cleavage, SaCas9 autonomously releases the PAM-distal DNA while retaining binding to the PAM. This partial DNA release immediately abolishes its strong interaction with the protospacer DNA and consequently promotes its subsequent dissociation from the PAM. Overall, these data provide a dynamic understanding of SaCas9 and instruct its effective applications.

**Keywords** CRISPR; DNA-protein interaction; Helicase; SaCas9; single molecule
**Subject Categories** Chromatin, Transcription, & Genomics

## Introduction

Clustered regularly interspaced short palindromic repeats (CRISPR)-associated (Cas) systems equip bacteria and archaea with adaptive immunity against foreign invasive genetic elements (Deveau *et al*, 2010; Marraffini & Sontheimer, 2010). In type II CRISPR-Cas systems, a single Cas9 protein in complex with a dual-guide RNA comprising CRISPR RNA (crRNA) and trans-activating RNA (tracrRNA) is sufficient to identify and cleave foreign DNA (Jinek *et al*, 2012). This ribonucleoprotein complex functions as a programmable endonuclease that targets complementary DNA sequences that are immediately close to a protospacer adjacent motif (PAM) for site-specific double-stranded DNA (dsDNA) cleavage. The simplicity, flexibility, and high efficiency of these systems have given rise to their wide in vivo applications as versatile genome tools in various organisms, such as in genome editing, transcription modulation, and live-cell imagining of chromosomal loci (Mali *et al*, 2013; Ma *et al*, 2016; Knott & Doudna, 2018). However, the in vivo delivery of the Cas9 enzyme with engineered single-guide RNA (sgRNA, which is a synthetic fusion of crRNA and tracrRNA) is often restricted by the cargo sizes of viral vectors (Jinek *et al*, 2012). Compared with the commonly used *Streptococcus pyogenes* Cas9 (SpCas9) which has 1,368 amino acids, one of its smallest orthologues, *Staphylococcus aureus* Cas9 (SaCas9), is a 1,053 amino acid enzyme with only a 17% sequence similarity to SpCas9. However, it shares very similar core folds with SpCas9 and exhibits a comparable genome editing efficiency in vivo (Nishimasu *et al*, 2014, 2015), thus providing a promising platform for therapeutic applications. Advances in our understanding of the molecular mechanism of SaCas9 underlying its RNA-guided DNA recognition and cleavage would not only facilitate its better usage as a genome tool but also aid in the development of its derivatives and the expansion of its applications.

Tremendous efforts have been made to understand the molecular mechanisms of Cas proteins (Cuculis & Schroeder, 2017; Jiang & Doudna, 2017). Specifically, biochemical, structural and single-molecule studies have been extensively conducted to characterize the detailed molecular mechanism of SpCas9, which could provide insights into our understanding of SaCas9 (Nishimasu *et al*, 2014; Sternberg *et al*, 2014; Szczelkun *et al*, 2014; Knight *et al*, 2015; Chen *et al*, 2017; Singh *et al*, 2018; Newton *et al*, 2019; Ivanov

1 School of Life Science and Technology, ShanghaiTech University, Shanghai, China
2 Shanghai Institute of Biochemistry and Cell Biology, Chinese Academy of Sciences, Shanghai, China
3 University of Chinese Academy of Sciences, Beijing, China
4 College of Life Sciences, Northwest A&F University, Yangling, China
5 State Key Laboratory of Reproductive Medicine, Center for Global Health, Nanjing Medical University, Nanjing, China
6 The LBPA, Ecole Normale Supérieure Paris-Saclay, CNRS, Université Paris Saclay, Gif-sur-Yvette, France
*Corresponding author. Tel: +86 21 2068 4536; E-mail: sunbo@shanghaitech.edu.cn
†These authors contributed equally to this work

*et al*, 2020). Upon being complexed with sgRNA, SpCas9 undergoes a significant structural rearrangement and becomes ready to recognize a 5'-NGG-3' PAM on the nontarget strand (Fonfara *et al*, 2013). This recognition triggers directional protospacer DNA unwinding, RNA–DNA hybridization, and R-loop expansion, which initiates from the PAM-proximal region (Sternberg *et al*, 2014). R-loop formation drives the repositioning and reorientation of one of its endonuclease domains (HNH) to a position in which it can access the cleavage site on the target strand (Nishimasu *et al*, 2014; Sternberg *et al*, 2015; Dagdas *et al*, 2017; Huai *et al*, 2017; Yang *et al*, 2018; Zuo & Liu, 2019). Intriguingly, both in vitro and in vivo studies demonstrated that SpCas9 stably binds to both cleaved DNA ends for hours without dissociation, resulting in the enzyme being a single-turnover nuclease (Sternberg *et al*, 2014; Ma *et al*, 2016; Jones *et al*, 2017). This characteristic limits the usage of each SpCas9 protein and impedes subsequent DNA repair after a DNA double-strand break (DSB). On the other hand, SaCas9 recognizes a unique 5'-NNGRRT-3' PAM sequence and requires a specific 20–22 nt DNA sequence (Friedland *et al*, 2015; Ran *et al*, 2015; Xie *et al*, 2018). Efforts have been made to improve and broaden its PAM specificities (Kleinstiver *et al*, 2015; Tan *et al*, 2019). Interestingly, in contrast with SpCas9, SaCas9 has been shown to function as a multiple-turnover enzyme (Yourik *et al*, 2019), indicating subtle mechanistic differences between these two proteins. Although the static structure of SaCas9 in complex with sgRNA and target DNA has provided valuable insights into the molecular basis of its function (Nishimasu *et al*, 2015), the dynamics of SaCas9 in its interplay with the DNA target remain unknown.

Herein, using single-molecule and ensemble approaches, we thoroughly characterized the molecular mechanism of SaCas9 in DNA target association and dissociation. Analyses of SaCas9 with partially matched DNA targets revealed that its DNA association requires 6 bp RNA–DNA matches close to the PAM, whereas efficient DNA unwinding and cleavage by SaCas9 require at least 18 bp PAM-proximal matches. Moreover, two conserved interaction sites between SaCas9 and its DNA target have been identified. Notably, one of them within the protospacer is too strong to be effectively disrupted by DNA-tracking motors and lasts until DNA cleavage. In addition, SaCas9 differs from SpCas9 in its autonomous release of the PAM-distal DNA after cleavage. Overall, our data enable a direct mechanistic comparison between SaCas9 and SpCas9, revealing notable differences in their DNA target binding, unwinding, cleavage, and dissociation. The distinct characteristics of SaCas9 presented in this work advance our understanding of its RNA-guided DNA cleavage mechanism and may instruct its effective in vivo applications as a genome tool.

# Results

## Probing positions and strengths of dSaCas9-DNA interactions

The association of SaCas9 with its DNA target is dictated by critical and specific interactions between them. We first sought to quantitatively determine the positions and strengths of these interactions. To do so, we adopted a previously developed optical tweezer-based DNA unzipping technique (Koch *et al*, 2002; Koch & Wang, 2003; Hall *et al*, 2009). In this assay, a DNA molecule with a three-way junction formed by two arms and a trunk was used (Fig EV1A–C). The ends of the two arms were attached to the surface of a microscope coverslip and to a microsphere held in an optical trap. A single SaCas9-sgRNA complex was uniquely positioned at a specially designed sequence on the trunk. Mechanically unzipping the trunk leads to the disruption of interactions between SaCas9 and DNA at well-defined locations with distinct forces, and the magnitudes of which reflect the strengths of these interactions. Unless otherwise stated, the DNA unzipping was carried out at a rate of 50 nm/s which resembles the motion rate of a typical DNA helicase (Manosas *et al*, 2010).

We first utilized a catalytically dead version of SaCas9 (hereafter referred to as dSaCas9) with a predesigned sgRNA which was fully matched with the target DNA (sgRNA-1, Table EV1; Ran *et al*, 2015). The 22-bp matched DNA sequence from the PAM-distal side to the PAM-proximal side was defined as +22 to +1, and the PAM sequence was sequentially designated as 0 to −5 for the sake of data presentation (Fig 1A). In this experiment, after forming DNA tethers in a chamber, we flowed dSaCas9 complexed with sgRNA into it and incubated the mixture for 10 min, followed by washing the unbound proteins and sgRNAs out of the chamber to avoid off-target binding (all subsequent unzipping experiments were performed after the washout, unless otherwise stated). By initiating DNA unzipping from the upstream side of the PAM (termed forward unzipping; Fig 1A), we found that the detected forces rose above those of the naked DNA baseline exclusively at the expected dSaCas9-binding position (Fig 1B). Control experiments with either sgRNA or dSaCas9 exhibited no such rise in force (Fig EV1D and E), suggesting that the observed single force increase is attributable to the disruption of a stable interaction site formed by the dSaCas9/

▶

---

**Figure 1.  The interaction sites and strengths between dSaCas9 and DNA.**

A   Cartoon illustrating the single-molecule DNA unzipping experiment used for the detection of the interactions between dSaCas9 and DNA and the definition of the DNA sequence.

B   Representative traces of the forward DNA unzipping in the presence of dSaCas9/sgRNA showing the force versus the number of unzipped base pairs. The naked DNA unzipping signatures are also presented for comparison (gray). Insets: zoomed-in view of the regions with increases in force. The dashed lines illustrate the interaction sites between dSaCas9 and the DNA. Black arrows indicate the force peaks of the pre-PAM interaction.

C   Representative traces of the reverse DNA unzipping in the presence of dSaCas9/sgRNA showing the force versus the number of unzipped base pairs. Insets: zoom-in view of the regions with increases in force. The dashed lines illustrate the interaction sites between dSaCas9 and the DNA. Black and blue arrows indicate the force peaks of the pre-PAM and post-PAM interactions, respectively.

D   Positions and strengths of the two interactions between dSaCas9 and DNA. The black and blue triangles indicate the pre- and post-PAM interactions, respectively. In total, 27 and 34 traces were collected in the forward and reverse unzipping assays, respectively. pre-PAM interactions (black) were detected in both the forward and reverse DNA unzipping assays. The error bars represent the SD.

E   Exonuclease III footprinting of dSaCas9 on the DNA target with Cy5 labeled on the 5' end of the nontarget strand.

F   Exonuclease III footprinting of dSaCas9 on the DNA target with Cy5 labeled on the 5' end of the target strand.

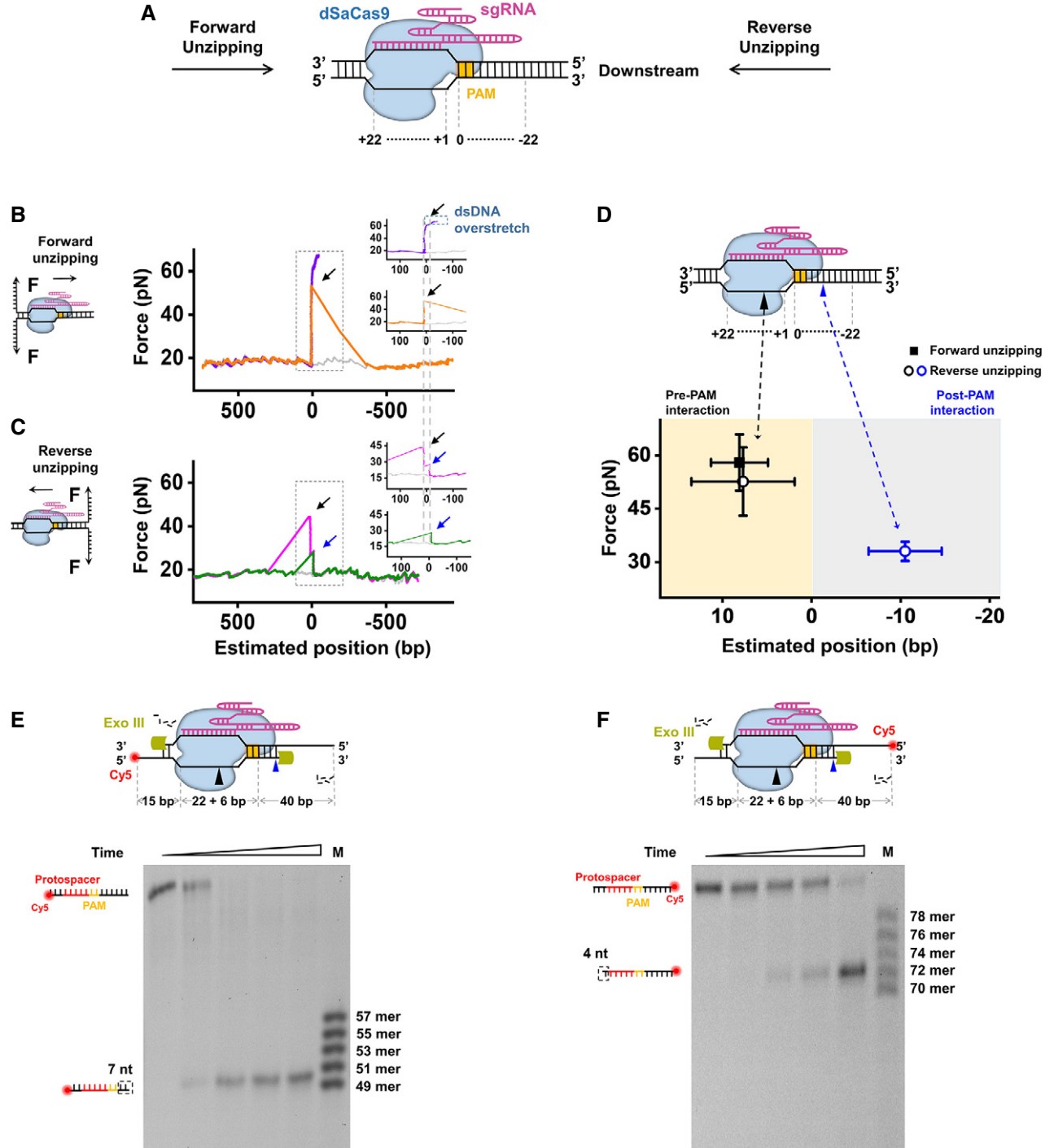

Figure 1.

sgRNA/DNA complex. Based on the assumptions that the protospacer was still intact dsDNA upon binding by dSaCas9 and that one separated base pair generated two nucleotides of ssDNA, this interaction site is estimated to be located 8.1 ± 3.2 bp (mean ± SD) away from to the PAM (Figs 1D and EV2A), and within the protospacer region; thus, it is termed the pre-PAM interaction. Careful analysis of the strength of this interaction site revealed two types of unzipping signatures. In 30% of the examined traces, the forces

dropped to ~ 17 pN after the rise and continued to be similar to those of the corresponding naked DNA (Fig 1B, orange trace), indicating that the ternary complex was disrupted. However, in the remaining traces, the complex presented a strong resistance that could not be disrupted even after the force went above 65 pN, under which the dsDNA arms underwent overstretching (Fig 1B, purple trace) (Smith et al, 1996). The magnitude of this single rise in force averaged 58.0 ± 7.9 pN (mean ± SD; traces showing dsDNA

overstretching were counted as a 65 pN disruption force in the statistics; Figs 1D and EV2B) and were comparable under a wide range of unzipping speeds (Fig EV2C). These data exhibit an extremely high affinity of dSaCas9 for the DNA target.

Next, we unzipped the DNA molecule with a preassociated dSaCas9 from the downstream side of the PAM (termed reverse unzipping; Fig 1A). After comparing the resulting force signal with that of the naked DNA, two types of unzipping traces were immediately noticed (Fig 1C). Among the 34 examined traces, 18 showed a single force peak (Fig 1C, green trace) at $33.0 \pm 2.7$ pN (mean $\pm$ SD) with an average position of $-10.4 \pm 4.4$ bp (mean $\pm$ SD; Figs 1D and EV2D–F). This result indicates an unexpected interaction site approximately 6 bp downstream of the PAM (termed the post-PAM interaction), which is beyond the apparent footprint of SaCas9 on the target DNA. Moreover, the fact that the relatively stronger pre-PAM interaction site was not detected after the disruption of the post-PAM interaction in these traces suggests an immediate collapse of the ternary complex. However, the remaining traces presented two peaks at the expected dSaCas9-binding position (Fig 1C, pink trace). Following the detection of this post-PAM interaction, an additional peak at $7.7 \pm 5.8$ bp (mean $\pm$ SD) with an average of $52.6 \pm 9.6$ pN (mean $\pm$ SD) was subsequently recorded (Figs 1D and EV2D–F). This detected interaction falls into the same category as the disruption force detected in the forward unzipping assay in terms of both position and strength, making it attributable to the pre-PAM interaction. Experiments with another sgRNA (sgRNA-2, Table EV1) produced similar results (Appendix Fig S1); thus, it is highly unlikely that these detected interactions are sequence-dependent. In addition, we also found that the lifetime of a DNA-bound dSaCas9 is longer than 24 h (Fig EV2G).

The existence of the post-PAM interaction implies that the footprint of dSaCas9 on the target DNA should be wider than the regions of the protospacer and the PAM. To test this hypothesis, we carried out DNA footprinting assays with near single-nucleotide resolution. A denaturing PAGE analysis of the products of cleavage of 83 bp dsDNA (including a 22-bp protospacer DNA and a 6-bp PAM, Table EV1) by Exonuclease III showed that 7 bp DNA downstream of the PAM and 4 bp beyond the protospacer remained after prebinding by dSaCas9, resulting in its footprint of 39 bp on the target DNA (Fig 1E and F). These results are consistent with our DNA unzipping data and further corroborate the existence of the post-PAM interaction.

Taken together, these results indicate that two stable interactions flanking the PAM exist between dSaCas9 and its DNA target.

## dSaCas9 presents a strong barrier to DNA-tracking motors

We recently reported similar interaction sites of SpCas9 with its DNA target (Zhang *et al*, 2019). However, dSaCas9 differs from dSpCas9 in that its pre-PAM interaction is significantly strong and is not affected by disruption from the post-PAM interaction in some cases (Fig 1C). This difference implies that dSaCas9 could act as a more effective DNA roadblock compared to dSpCas9 to perturb DNA-based transactions, such as DNA replication. To test this hypothesis, we first chose to use DnaB (in complex with DnaC, hereafter referred to as DnaB), which is an *E. coli* replicative helicase that catalyzes the separation of dsDNA during DNA replication

(LeBowitz & McMacken, 1986), to mimic the replication fork and examine the outcome of its encountering with a DNA-bound dSaCas9. A previously developed single-molecule optical tweezer assay was employed (Sun *et al*, 2011, 2015, 2018). The helicase-catalyzed unwinding of a DNA fork junction was monitored via the increase in the ssDNA length under 12 pN, which was not sufficient to mechanically unzip the fork junction. In the absence of dSaCas9, the DnaB helicase was found to smoothly unwind naked DNA without obvious pauses from both the upstream and downstream sides of the PAM (Fig 2A and B). However, once prebound by dSaCas9, the unwinding of the DnaB helicase from both directions was completely stalled at the expected dSaCas9-binding position within 200 s (our experimental cutoff time; Fig 2A and B). Consistently, the stably bound dSaCas9 also repelled the BLM-core helicase (BLM[642–1290]), which is a typical homologous recombination-associated helicase (Janscak *et al*, 2003), from both directions (Fig 2C and D). The slight decrease in the DNA length upon the collision can be explained by BLM switching strands and translocating onto the opposite ssDNA upon encountering dSaCas9 (Wu, 2007; Wang *et al*, 2015). Moreover, Phi29 DNA polymerase (DNAP), which is a strong DNA-based motor that can perform strand-displacement replication at a fast rate alone (Morin *et al*, 2012), was also found to be incapable of displacing DNA-bound dSaCas9 proteins in our strand-displacement replication assay (Fig 2E and F; Sun *et al*, 2015). Instead, as indicated by the decreases in the DNA length upon the collisions, the exonuclease activity of Phi29 DNAP was more prone to being activated. It is noteworthy that these results are in stark contrast to our previous findings with dSpCas9, in which it could be disrupted by BLM and Phi29 DNAP from both directions (Zhang *et al*, 2019).

Overall, we conclude that the potent association of dSaCas9 with its DNA target poses it a strong barrier to DNA-tracking motors.

## Effects of mismatches on SaCas9 activity

Off-target binding and cleavage of Cas9 proteins often occur at partially matched DNA sites (Fu *et al*, 2013). We next aimed to address how mismatches between DNA targets and sgRNA affect the DNA association and cleavage of SaCas9. To this end, we first analyzed the abilities of a series of sgRNAs harboring consecutive double-nucleotide mismatches to guide wild-type SaCas9 (hereafter referred to as SaCas9) for DNA cleavage (Fig 3). We found that the cleavage efficiency of SaCas9 showed different mismatch sensitivities at different RNA–DNA hybrid regions. Notably, PAM-proximal mismatches at positions 1–6 can result in deficient DNA cleavage by SaCas9, highlighting the importance of the complementarity of the PAM-proximal DNA with sgRNA. This finding is consistent with a previous study showing that RNA–DNA base pairing in the PAM-proximal "seed" region is critical for SaCas9-catalyzed DNA cleavage (Tycko *et al*, 2018).

Given the importance of the PAM-proximal "seed" region, we next introduced mismatches at the PAM-distal end (denoted using the naming convention $x\text{-}y_{MM}$ where the *x*th through *y*th bps are mismatched) to examine their effects on DNA binding, unwinding, and cleavage by SaCas9 (Fig 4A). We first repeated the reverse DNA unzipping assay using DNA templates with different matching degrees relative to sgRNA-1 (Table EV1). Traces with only four or five matches showed the absence of stable DNA binding of dSaCas9

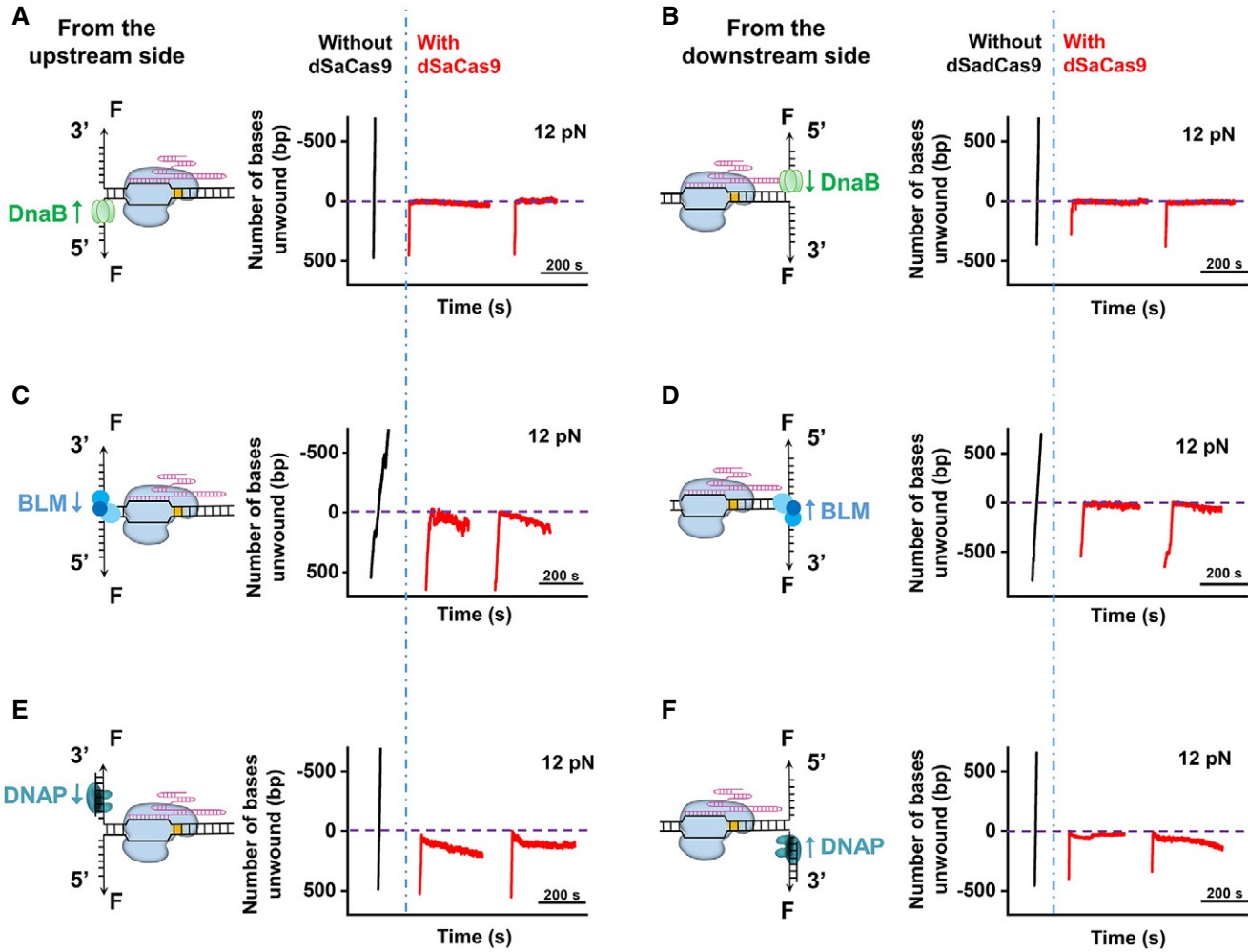

**Figure 2. DNA-tracking motors were stalled by DNA-bound dSaCas9.**

A   DNA unwinding by DnaB was initiated from the upstream side of the PAM in the presence of dSaCas9 ($n = 10$). Representative traces show the number of unwound base pairs versus the time under an assisting force of 12 pN in the absence (black) or presence (red) of the prebound dSaCas9. For clarity, the traces have been shifted along the time axis. The dashed lines indicate the expected dSaCas9-binding positions.

B   DNA unwinding by DnaB was initiated from the downstream side of the PAM in the presence of dSaCas9 ($n = 11$). Representative traces show the number of unwound base pairs versus the time under an assisting force of 12 pN in the absence (black) or presence (red) of the prebound dSaCas9.

C   DNA unwinding by BLM was initiated from the upstream side of the PAM in the presence of dSaCas9 ($n = 12$). Representative traces show the number of unwound base pairs versus the time under an assisting force of 12 pN in the absence (black) or presence (red) of the prebound dSaCas9.

D   DNA unwinding by BLM was initiated from the downstream side of the PAM in the presence of dSaCas9 ($n = 14$). Representative traces show the number of unwound base pairs versus the time under an assisting force of 12 pN in the absence (black) or presence (red) of the prebound dSaCas9.

E   Phi29 DNAP strand-displacement synthesis was initiated from the upstream side of the PAM ($n = 15$). Representative traces show the number of unwound/synthesized base pairs versus the time under an assisting force of 12 pN in the absence (black) or presence (red) of the prebound dSaCas9.

F   Phi29 DNAP strand-displacement synthesis was initiated from the downstream side of the PAM ($n = 19$). Representative traces show the number of unwound/synthesized base pairs versus the time under an assisting force of 12 pN in the absence (black) or presence (red) of the prebound dSaCas9.

(Figs 4A and EV3A). However, once 6 or more RNA–DNA matches close to the PAM existed, nearly all examined traces were found to associate with the dSaCas9 protein at the mismatched positions (Figs 4A and EV3A). Similar results were obtained from the forward DNA unzipping assays using these DNA templates (Fig EV3B). Additionally, once the matching number reached 14 bp, traces from the reverse unzipping assays started to show two peaks in force at the binding sites, and the percentage of two-peak traces increased with the increase in the matching number (Figs 4A and EV3A). These

findings suggest that DNA binding of dSaCas9 becomes more stable with the increase in RNA–DNA matches. Analysis of the unzipping signatures revealed that two interactions mediate dSaCas9 binding with imperfect RNA/DNA complementarity, resembling those of the fully matched DNA target (Figs 1D and 4B). These interactions were further confirmed by experiments in which mismatches were introduced using a series of sgRNAs (Appendix Fig S2).

Upon PAM recognition, the Cas9 protein commonly initiates PAM-proximal DNA unwinding and expands an R-loop to examine

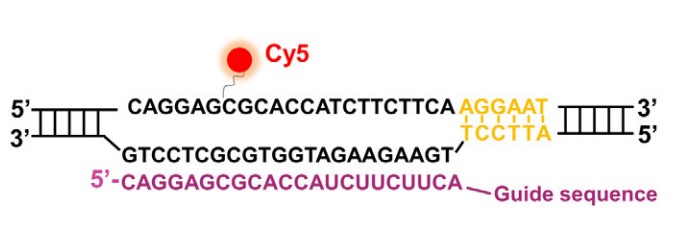

**RNA sequences**

1-2_MM    5'- CAGGAGCGCACCAUCUUCUUUG -3'

3-4_MM    5'- CAGGAGCGCACCAUCUUCCCCA -3'

5-6_MM    5'- CAGGAGCGCACCAUCUCUUUCA -3'

7-8_MM    5'- CAGGAGCGCACCAUUCUCUUCA -3'

9-10_MM   5'- CAGGAGCGCACCGCCUUCUUCA -3'

11-12_MM  5'- CAGGAGCGCAUUAUCUUCUUCA -3'

13-14_MM  5'- CAGGAGCGUGCCAUCUUCUUCA -3'

15-16_MM  5'- CAGGAGUACACCAUCUUCUUCA -3'

17-18_MM  5'- CAGGGACGCACCAUCUUCUUCA -3'

19-20_MM  5'- CAAAAGCGCACCAUCUUCUUCA -3'

21-22_MM  5'- UGGGAGCGCACCAUCUUCUUCA -3'

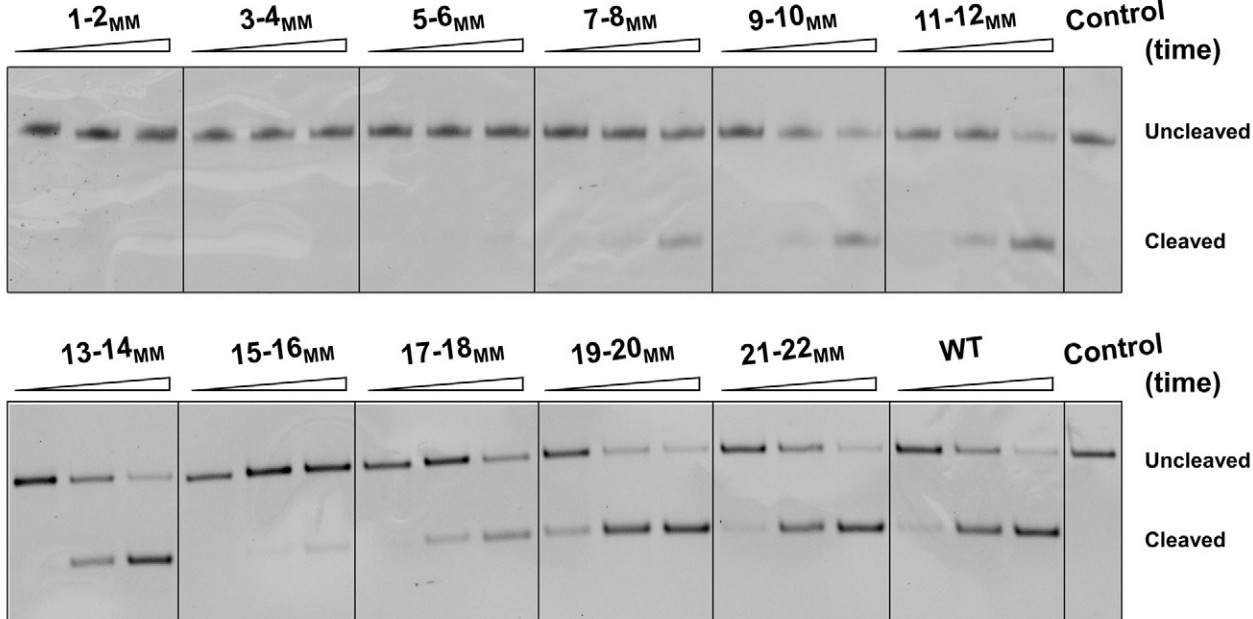

**Figure 3.   DNA cleavage by SaCas9 with sgRNAs harboring consecutive double-nucleotide mismatches.**
Schematic representation of the sequences of the DNA target and sgRNAs. The PAM is shown in yellow. The nontarget DNA strand was labeled with Cy5. The matched and mismatched sequences in sgRNAs are shown in purple and blue, respectively. Representative gel shows DNA cleavage by SaCas9 guided by partially matched sgRNAs.

the RNA/DNA complementarity before cleavage. We next employed a stopped-flow assay to detect DNA unwinding by dSaCas9 and examine its dependence on the RNA/DNA complementarity. We used a DNA substrate that includes a 2-aminopurine (2-AP) at position 18 of the nontarget strand to report the late stage of R-loop expansion (Fig 4C). Since the 2-AP fluorescence intensity is quenched by base stacking but not in a single-stranded state, unwinding of the protospacer DNA by dSaCas9 would result in an increase in the fluorescence (Gong *et al*, 2018; Strohkendl *et al*, 2018). With the presence of dSaCas9 and a fully matched sgRNA, the 2-AP fluorescence signal indeed showed increase in fluorescence, with rates of 5.5 and 1.1 s$^{-1}$ when fitted to a double-exponential equation (Fig 4C and D). This result suggests two sequential

unwinding events. A kinetic study of SpCas9 also showed two unwinding events and suggested that they are related to HNH cleavage and RuvC cleavage (Gong *et al*, 2018). Using a 19–22_MM sgRNA, we observed a similar increase in fluorescence, with rates of 5.1 and 1.1 s$^{-1}$ (Fig 4C and D). However, experiments with 6 or more mismatched sgRNAs showed an additional slow decrease in fluorescence after the increase, indicating an unstable R-loop formation (Fig 4C and D). These findings indicate that complete R-loop formation by dSaCas9 requires 18 RNA–DNA matches and occurs within one second. We also conducted DNA cleavage assays with mismatched sgRNAs and found that DNA cleavage products were not observed until there were 18 bp PAM-proximal matches (Figs 4E and EV3C). This result was further confirmed by

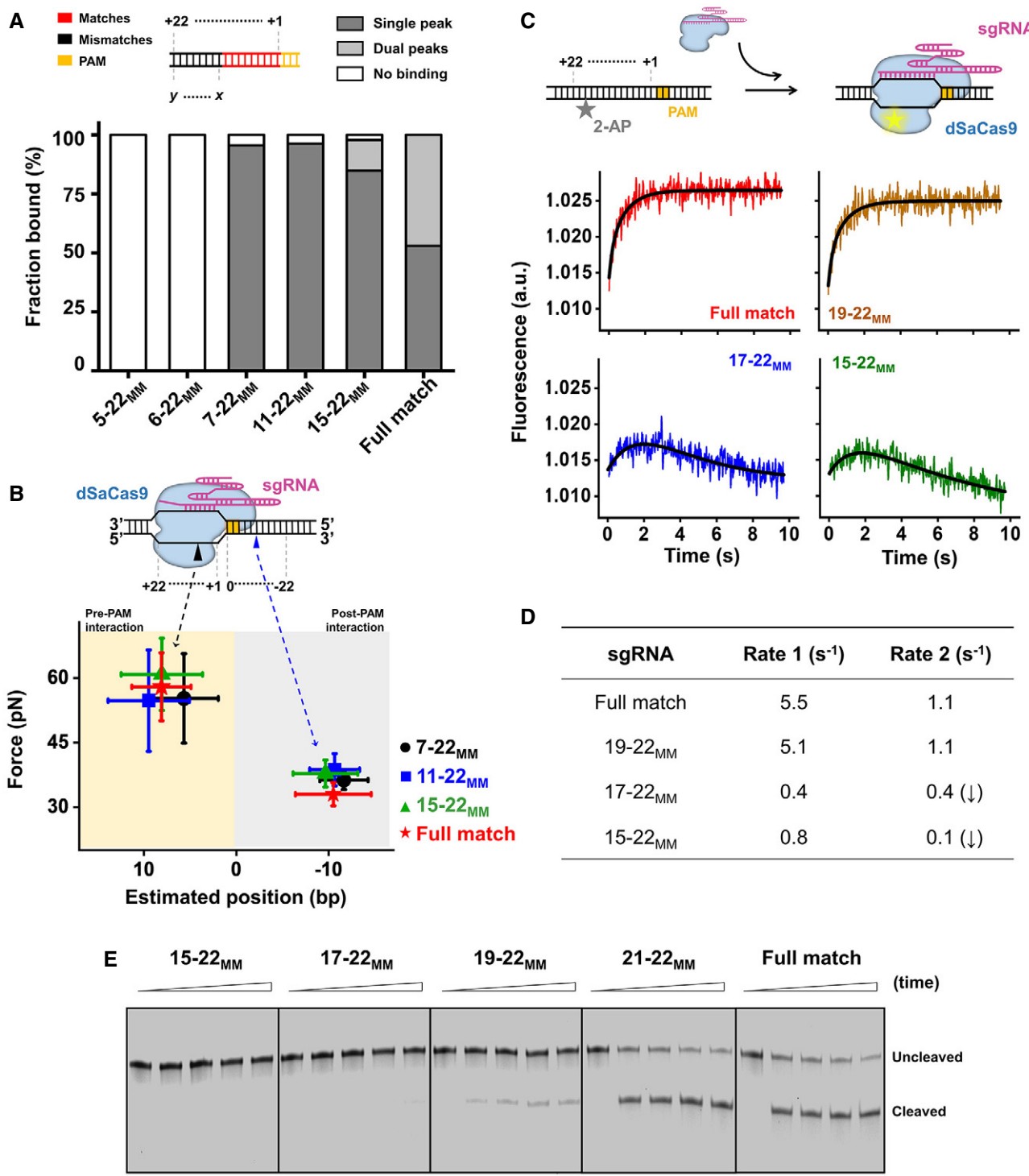

**Figure 4. Effects of mismatches on SaCas9 activity.**

A    DNA-binding fractions by dSaCas9 at the fully matched (target) and partially matched DNA target positions, as revealed by the reverse DNA unzipping assays ($n$ = 15, 16, 22, 26, 46, and 34 from left to right). The subscript "MM" represents the mismatches of the DNA sequence with the sgRNAs.

B    Positions and strengths of the pre-PAM interactions ($n_{7–22MM}$ = 21, $n_{11–22MM}$ = 41, $n_{15–22MM}$ = 39) and post-PAM interactions ($n_{7–22MM}$ = 21, $n_{11–22MM}$ = 25, $n_{15–22MM}$ = 45) between dSaCas9/sgRNA and mismatched DNA targets. The error bars represent the SD.

C    Schematic depiction of the stopped-flow experiment. 2-AP was substituted at position + 18 of the nontarget strand. The 2-AP fluorescence signals as a function of time are shown for four sgRNAs. The solid lines are double-exponential fits of the data.

D    A summary of fitted values of rates.

E    Cleavage activities of SaCas9 on DNA targets with sgRNA containing bases mismatched to DNA. The reactions were quenched at five time points (0, 15, 30, 45, and 60 min). The experiment was performed in triplicate, and a representative gel is shown.

experiments using another set of DNA template and sgRNAs (Appendix S3).

Overall, we conclude that the stable DNA binding of SaCas9 only requires RNA–DNA pairing in the 6 bp PAM-proximal region, and efficient DNA unwinding and cleavage by SaCas9 requires up to 18 bp matches.

### SaCas9 autonomously releases the PAM-distal DNA after cleavage

A previous study showed that unlike SpCas9, SaCas9 can act as a multiple-turnover endonuclease, indicating relatively quick dissociation of SaCas9 from the DNA after cleavage (Yourik *et al*, 2019). To examine how SaCas9 dissociates from the DNA, we performed forward and reverse DNA unzipping assays with wild-type SaCas9. In the absence of $Mg^{2+}$, we detected two stable interactions between SaCas9 and DNA that resemble those of dSaCas9 in terms of positions and strengths (Figs 1D and EV4A). In the presence of $Mg^{2+}$, reverse DNA unzipping experiments performed within 2 h always showed an exclusive rise in force at the expected SaCas9-binding position; however, after the rise, the unzipping force immediately dropped to zero (Fig 5A). We reasoned that the lack of a naked DNA unzipping signature after the rise in force was due to the breakage of the dsDNA induced by the endonucleolytic activity of SaCas9 (Fig 5A). This detected interaction is located $-12.9 \pm 3.1$ bp (mean $\pm$ SD) from the PAM and resembles the post-PAM interaction found with dSaCas9 (Figs 1D and EV4A). It averaged $49.4 \pm 10.5$ pN (mean $\pm$ SD) and was slightly stronger than that of dSaCas9, possibly resulting from a stabilizing effect of $Mg^{2+}$ (Figs 1D and EV4A). These data strongly suggest that SaCas9 still binds to the PAM region after DNA cleavage. However, in contrast to the reverse DNA unzipping, up to 70% of traces in the forward DNA unzipping assay showed that the force directly dropped to zero near the expected SaCas9-binding position (Fig 5B). There are two explanations accounting for the observed force-drop. One possibility is that SaCas9 could autonomously release the PAM-distal DNA after cleavage, resulting in the loss of detection of the rise in force. Alternatively, DNA-cleaved SaCas9 could still bind to the PAM-distal DNA, wherein, after cleavage, this binding is too weak to be detected by our DNA unzipping assay. We next employed a combination of optical tweezers and confocal microscopy to differentiate these two possibilities. In this assay, an individual TO-PRO-3-labeled λ DNA molecule was suspended between two microspheres held in two traps and fluorescently monitored, and its associated SaCas9 in complex with a 5'-Cy3-labeled crRNA and an unlabeled tracrRNA was simultaneously monitored (Fig 5C and Table EV1). The downstream and upstream regions of the on-target sequence on the λ DNA molecule are 36 and 12 kbp long, respectively. In the presence of the dSaCas9 and crRNA:tracrRNA complex, we observed a single binding event at the expected location on the suspended λ DNA (Fig 5C). Control experiments verified that this observation represents a stable binding event of the complex (Appendix Fig S4). However, the substitution of dSaCas9 by wild-type SaCas9 resulted in an observation of a flow-stretched DNA segment, with the complex bound to one of its ends (Fig 5C). This stretched DNA segment was typically approximately 36 kbp, indicating a loss of the 12 kbp DNA segment after cleavage by SaCas9. These data favor the explanation that SaCas9 autonomously

releases the PAM-distal DNA while remaining associated with the PAM. The partial DNA release mechanism of SaCas9 is in stark contrast with SpCas9, which firmly binds to both DNA ends after cleavage (Nishimasu *et al*, 2014; Sternberg *et al*, 2014).

How does a PAM-bound SaCas9 dissociate after releasing the PAM-distal DNA? We first examined the lifetime of a PAM-bound SaCas9 and found that SaCas9 can indeed autonomously dissociate from the PAM within several hours after cleavage, whereas SpCas9 remains associated with the PAM over 24 h (Fig EV4B and C). Since the autonomous release of the PAM-distal DNA immediately abolishes the strong pre-PAM interaction of SaCas9 (Fig 5A and B), it might be possible for DNA-tracking motors to disrupt the remaining post-PAM interaction of SaCas9 and to facilitate its final dissociation. We thus examined the outcomes of collisions of downstream helicases or DNAP with prebound SaCas9. The Phi29 DNAP and DnaB were still found to stall at the expected SaCas9-binding position for up to 200 s (Fig 5D and E). However, the DNA repair helicase BLM was indeed able to disrupt the SaCas9 protein from the downstream side of the PAM, as evident from the breakage of the DNA tether when the two proteins met (Fig 5F). These data indicate that DNA repair motors may facilitate the dissociation of a PAM-bound SaCas9 for subsequent repair.

## Discussion

In this work, we quantitatively determined the locations of SaCas9-DNA interaction sites at a few-base-pair resolution and their corresponding strengths. Two dominant interactions between dSaCas9 and its target DNA have been recognized, with one within the protospacer (pre-PAM) and the other ~ 6 bp downstream of the PAM (post-PAM; Fig 1B–D and Appendix S1). According to the crystal structure of DNA-bound SaCas9 (Nishimasu *et al*, 2015), the post-PAM interaction site is beyond the apparent footprint of SaCas9 on the target DNA, illustrating additional DNA coverage. Our subsequent DNA footprinting assays further corroborated this broadened coverage of dSaCas9 on the DNA target (Fig 1E and F). Both structural and single-molecule studies with SpCas9 also revealed the existence of a similar post-PAM interaction (Huai *et al*, 2017; Zhang *et al*, 2019). This conserved feature highlights the importance of this unexpected interaction site for Cas9 proteins. Since it is located close to the PAM, it is tempting to speculate that Cas9 proteins may use this interaction to help recognize the PAM and/or to modulate the DNA configuration, leading to the separation of the PAM-proximal DNA for R-loop formation. The other stable interaction between dSaCas9 and DNA is within the protospacer (Fig 1B and D). According to the structural data (Nishimasu *et al*, 2015), this interaction is possibly achieved via extensive interactions among the RNA–DNA heteroduplex close to the PAM, the bridge helix and the REC lobe of the SaCas9 protein. Compared with SpCas9, wherein the disruption of its post-PAM interaction results in an immediate loss of its pre-PAM interaction and consequently its dissociation from the DNA (Zhang *et al*, 2019), the pre-PAM interaction of dSaCas9 is significantly strong and can be less affected by the disruption of its post-PAM interaction (Fig 1C). Moreover, we demonstrated that even though the post-PAM interaction of wild-type SaCas9 is stronger than that of dSaCas9 (Figs 1D and EV4A), a downstream helicase can still displace it (Fig 5F). These findings

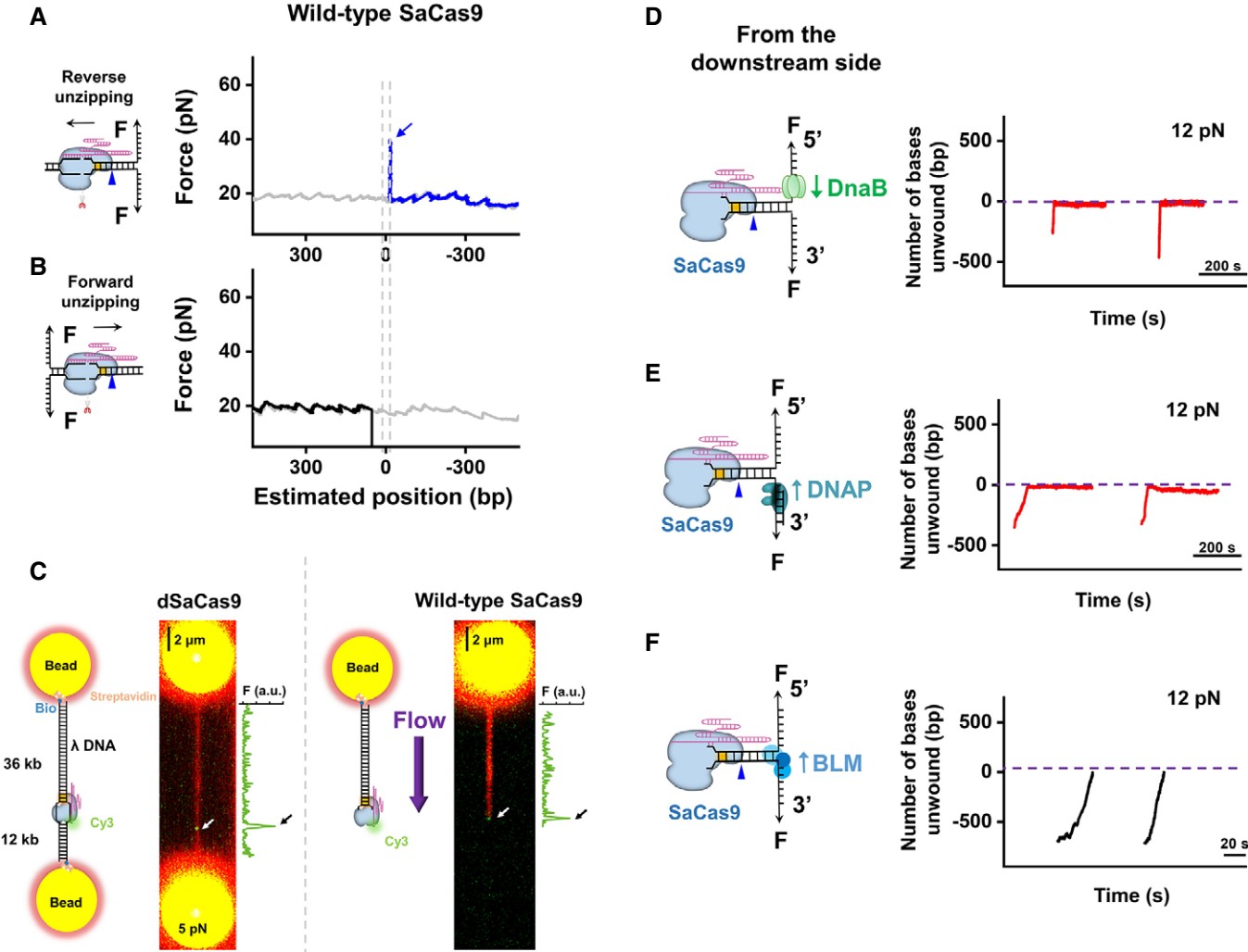

**Figure 5. The dissociation of SaCas9 after DNA cleavage.**

A Representative traces of the reverse DNA unzipping in the absence (gray) and presence (blue) of wild-type SaCas9 showing the force versus number of base pairs unzipped. In total, 23 traces were collected in this assay. The dashed line shows the expected interaction sites between dSaCas9 and DNA. The blue arrow indicates the force peak of the post-PAM interaction.

B Representative traces of the forward DNA unzipping in the absence (gray) and presence (black) of wild-type SaCas9 showing the force versus number of base pairs unzipped. In total, 49 traces were collected in this assay. The dashed line shows the expected interaction sites between dSaCas9 and DNA.

C Confocal images of fluorescently labeled λ DNA with dSaCas9 (n = 11) and wild-type SaCas9 (n = 16) bound to its DNA target. DNA-bound SaCas9 was visualized by labeling the 5′ end of the crRNA with Cy3. The λ DNA molecule was either suspended between two microspheres held by two optical traps under 5 pN or stretched by laminar flow. The fluorescence intensity of Cy3 alongside the image is shown to indicate the on-target binding of SaCas9.

D DNA unwinding by DnaB was initiated from the downstream side of the PAM after the association of the SaCas9 protein (n = 11). Representative traces show the number of unwound base pairs versus time under an assisting force of 12 pN. For clarity, the traces have been shifted along the time axis. The dotted lines indicate the expected SaCas9-binding positions.

E Phi29 DNAP-mediated strand-displacement synthesis was initiated from the downstream side of the PAM after the association of the SaCas9 protein (n = 13). Representative traces show the number of unwound/synthesized base pairs versus time under an assisting force of 12 pN.

F DNA unwinding by BLM was initiated from the downstream side of the PAM after the association of the SaCas9 protein (n = 23). Representative traces show the number of unwound base pairs versus time under an assisting force of 12 pN.

also suggest that the DNA-binding affinity of dSaCas9 is likely dominated by the pre-PAM interaction, and the disruption of the post-PAM interaction is not enough to dissociate the dSaCas9 from its target DNA. In addition to these two stable interaction sites, an additional intermittent interaction near the PAM-distal region was detected with SpCas9 (Zhang *et al*, 2019), whereas no such interaction was detected with SaCas9. Even though the structural data

showed that the C-terminal region of the REC lobe of SaCas9 interacts with the PAM-distal region of the heteroduplex (Nishimasu *et al*, 2015), it is possible that these interactions are too weak to be detected in our assays. The SaCas9-DNA interaction map reported herein has important implications for SaCas9 being used as a programmable DNA-binding protein to perturb the DNA metabolism and to locate DNA loci (Fig 2).

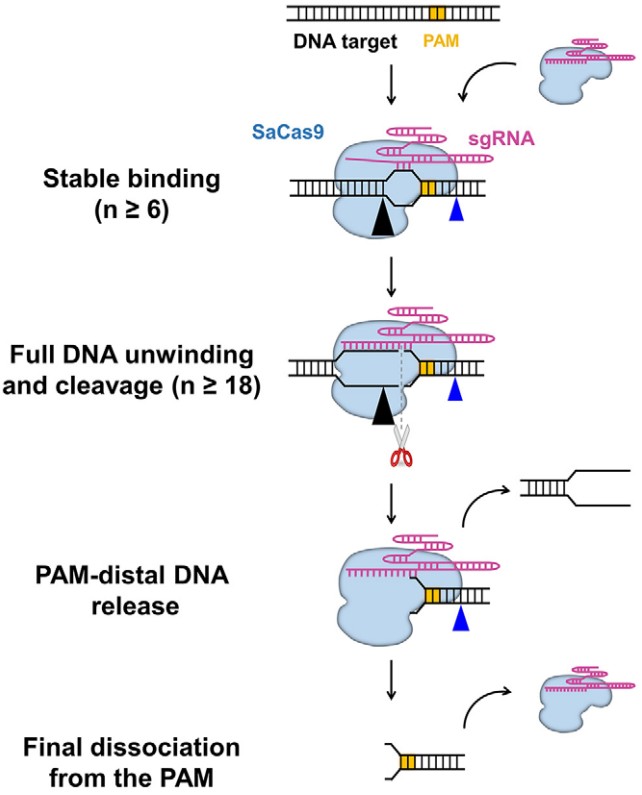

**Figure 6. Proposed model for SaCas9.**

Upon being complexed with sgRNA, SaCas9 confers stable binding to the DNA target when 6 PAM-proximal matches exist. It then triggers sequential DNA unwinding from the PAM-proximal region to the PAM-distal end and samples adjacent to the DNA for guide RNA complementarity. More than 18 bp RNA-DNA matches close to the PAM allow complete R-loop formation and endonucleolytic cleavage of both DNA strands. Afterward, SaCas9 remains bound to the PAM while autonomously releasing the PAM-distal DNA. The interplay between SaCas9 and the DNA is mediated by the post- and pre-PAM interactions, as indicated by the blue and black triangles, respectively.

This work, together with earlier data (Nishimasu *et al*, 2015; Ran *et al*, 2015; Tycko *et al*, 2018; Yourik *et al*, 2019), suggests a model for DNA target association and dissociation by SaCas9 that involves protein–DNA interactions at each stage of the reaction (Fig 6). The increased RNA–DNA matches required for SaCas9 to bind, unwind, and cleave DNA suggests stringent regulation of its endonucleolytic activity (Fig 4). Upon being complexed with the sgRNA, the stable DNA binding of SaCas9 is achieved by the separation of the 6-bp PAM-proximal protospacer DNA and its complementary with the sgRNA (Fig 4A and B). RNA–DNA hybridization initiates at the PAM-proximal region and the R-loop expands directionally to the PAM-distal region which is a process governed by the DNA-RNA matches (Fig 4C and D). Eighteen or more heteroduplex matches trigger SaCas9 endonuclease activity, which is followed by the release of the PAM-distal DNA (Figs 4E, 5B and 5C). This autonomous DNA release abolishes the pre-PAM interaction of SaCas9 with the DNA, leading to its final dissociation from the PAM (Figs 5 and EV4). A DNA repair helicase can accelerate the last step by directly displacing SaCas9 from the PAM (Fig 5F). Interestingly, the PAM-associated SaCas9 cannot be displaced by a DNA replicative helicase

or a polymerase (Fig 5D and E), suggesting that its displacement might be protein-dependent.

Our findings also allow us to make a direct mechanistic comparison between SpCas9 and SaCas9 in terms of their DNA target binding, unwinding, cleavage, and dissociation (Table 1). Although both structural data and our results suggest similarities between these two proteins (Nishimasu *et al*, 2014, 2015), SaCas9 is distinct from SpCas9 in a few mechanistic aspects besides the PAMs and their interaction sites with the DNA targets. The fact that 6 bp PAM-proximal matches are enough for SaCas9 to stable bind to DNA indicates that it is more tolerant to off-targeting binding sites compared to SpCas9, which needs 9 bp matches for stable binding (Fig 4A; Singh *et al*, 2016). However, unlike SpCas9 (Zeng *et al*, 2018), SaCas9 activity is more stringent with the first two PAM-proximal matches in terms of its endonuclease activity (Fig EV2). In addition to the subtly different association mechanisms, SaCas9 adopts a distinct dissociation mechanism by autonomously releasing the PAM-distal DNA after DNA cleavage (Fig 5), whereas SpCas9 binds to both DNA ends for hours without dissociation (Nishimasu *et al*, 2014; Sternberg *et al*, 2014; Knight *et al*, 2015; Newton *et al*, 2019). Compared with SpCas9, the relatively faster dissociation rate of SaCas9 allows it to be used as an effective multiple-turnover enzyme (Figs 5 and EV4; Yourik *et al*, 2019). Moreover, it also facilitates the subsequent repair of the DSB after cleavage (Fig 5F).

Overall, our findings have important implications for the use of SaCas9 as a genome engineering technology and might suggest new avenues for designing guide RNAs with improved specificities and developing Cas9 derivatives with increased efficiency.

## Materials and Methods

### Preparation of DNA templates and guide RNAs

The DNA template for the single-molecule unzipping assay consisted of three pieces, two arms, and a trunk and was prepared as previously described (Fig EV1B and C) (Sun & Wang, 2017). Briefly, arm 1 was PCR amplified from plasmid pBR322 using a digoxigenin-labeled primer and then digested with BstXI (NEB) to create an overhang. The resulting DNA fragment was subsequently annealed to a short DNA with a complementary overhang formed by adapters 1 and 2 (Table EV1). Arm 2 was PCR amplified from plasmid pBR322 using a biotin-labeled primer and was then digested with BstXI (NEB) to create an overhang. The DNA fragment was subsequently annealed to adapters 3 and 4 (Table EV1). Adapter 2 from arm 1 and adapter 4 from arm 2 were partially complementary to each other and were annealed to create a short 35-bp trunk with a 3-bp overhang for the trunk ligation. The 1,795 bp forward DNA unzipping trunk and 1,889 bp reverse DNA unzipping trunk containing the target DNA sequence were amplified from plasmid pEGFP-N1 and digested with AlwNI (NEB). The trunks containing DNA sequences partially matched to sgRNA were ligation products of upstream and downstream DNA segments with partially matched DNA segments that were annealed from two oligonucleotides (Table EV1 and Fig EV1C). The upstream DNA segment was digested with AlwNI (NEB) and BsaI (NEB), while the downstream segment was digested with

**Table 1. A mechanistic comparison between SpCas9 and SaCas9**

|  | SpCas9 | SaCas9 |
|---|---|---|
| Binding | $N \geq 9$ bp for stable binding;<br>post-PAM interaction: - 14 bp, 25 pN;<br>pre-PAM interaction: within the protospacer, 45 pN;<br>intermittent interaction: within the protospacer, 35 pN. | $N \geq 6$ bp for stable binding;<br>post-PAM interaction: - 6 bp, 33 pN;<br>pre-PAM interaction: within the protospacer, > 58 pN;<br>No detected intermittent interaction. |
| Unwinding | $N \geq 17$ bp for full protospacer unwinding;<br>Proceed from the PAM-proximal end to the PAM-distal end;<br>Rates: 1 s$^{-1}$ and 0.3 s$^{-1}$. | $N \geq 18$ bp for full protospacer unwinding;<br>Proceed from the PAM-proximal end to the PAM-distal end;<br>Rates: 5.5 s$^{-1}$ and 1.1 s$^{-1}$. |
| Cleavage | $N \geq 16$ bp; Not sensitive to 1–2 PAM-proximal mismatches. | $N \geq 18$ bp; Sensitive to 1–2 PAM-proximal mismatches. |
| Dissociation | Stably binds to both DNA ends after cleavage;<br>Molecular motors can readily displace it from both directions. | Autonomously releases PAM-distal DNA after cleavage;<br>DNA repair helicase can displace the PAM-bound SaCas9. |

$N$ represents the number of base pairings between the PAM-proximal DNA and the sgRNA. The post-PAM interaction sites for both proteins are shown without counting the PAM. SpCas9 data were obtained from references (Nishimasu et al, 2014; Sternberg et al, 2014; Knight et al, 2015; Singh et al, 2016, 2018; Gong et al, 2018; Zeng et al, 2018; Zhang et al, 2019).

BasI (NEB). The final product was produced by ligating the arms with the trunk at a 1:4 ratio.

The sgRNAs were transcribed from linearized pUC57-sgRNA expression vectors using the T7 High Efficiency Transcription Kit (Transgen Biotech). The sgRNAs were then purified by the EasyPure RNA Purification Kit (Transgen Biotech). The sequences of the sgRNAs are listed in Table EV1.

### Expression and purification of proteins

The dSaCas9 and wild-type SaCas9 proteins were purified as previously described (Nishimasu et al, 2015). Briefly, a synthetic gene coding for wild-type SaCas9 or dSaCas9 with an N-terminal His$_6$-tag was followed by a peptide sequence containing a tobacco etch virus (TEV) protease cleavage site. The sequence was synthesized and subcloned into pET24a to generate pET24a-Cas9. Proteins were expressed in the Escherichia coli strain BL21 Rosetta 2 (DE3) (Transgen Biotech) grown in LB at 37 °C for a few hours until the optical density at 600 nm reached 0.6, after which the culture temperature was lowered to 18 °C, and protein production was induced by the addition of 200 μM isopropyl β-D-1-thiogalactopyranoside (IPTG). The medium was then discarded, and the cells were harvested. The harvested cells were lysed in 20 mM Tris–HCl pH 8.0, 250 mM NaCl, 5 mM imidazole, and 1 mM phenylmethylsulfonyl fluoride (PMSF) and passed through a homogenizer three times at ~ 1,000 bar. The lysed dilution was then centrifuged at 13,000 g for 30–60 min, and the supernatant-clarified cell lysate was separated from the cellular debris and bound in batches to Ni-NTA agarose (Qiagen). The resin was washed extensively with 20 mM Tris–HCl pH 8.0, 250 mM NaCl, and 10 mM imidazole pH 8.0, and the bound protein was eluted in a single step with 20 mM Tris–HCl pH 8.0, 250 mM NaCl, and 250 mM imidazole. SaCas9 was dialyzed into dialysis buffer (20 mM HEPES-KOH pH 7.5, 150 mM KCl, 10% (v/v) glycerol, 1 mM dithiothreitol (DTT), and 1 mM EDTA) overnight at 4 °C. SaCas9 was further purified by a HiTrap SP HP Sepharose column (GE Healthcare) and by gel-filtration chromatography on a Superdex 200 16/60 column (GE Healthcare) in SaCas9 storage buffer (20 mM Tris–HCl, pH 7.5, 500 mM KCl, and 1 mM DTT) and was stored at −80 °C. The truncation mutant BLM$^{642-1290}$ (core-BLM) encompassing the region homologous to the RecQ catalytic core was purified as previously described (Janscak et al, 2003). DnaB and DnaC were purified as previously described (LeBowitz & McMacken, 1986). Phi29 DNAP was purchased from NEB (M0269S).

### Single-molecule DNA unzipping assays

DNA unzipping experiments were performed on an M-trap optical tweezer from LUMICKS (Amsterdam, Netherlands). Sample chamber preparation was similar to that previously described (Sun & Wang, 2017). Briefly, glass coverslips were cleaned and functionalized with partially biotinylated polyethylene glycol (PEG) (Laysan Bio; Yardimci et al, 2012) and then coated with streptavidin (Thermo). Biotin-tagged DNA was incubated to form DNA tethers. Anti-digoxigenin-coated 0.48 μm polystyrene microspheres (Polysciences) were added into the chamber. For single-molecule DNA unzipping experiments, wild-type SaCas9 or dSaCas9 was first complexed at 30 nM with a 1:3 ratio of protein to sgRNA at room temperature for 10 min and then flowed into the chamber just prior to data acquisition. The experiments were conducted in a climate-controlled room at a temperature of 23.3 °C; however, owing to local laser trap heating the temperature increased slightly to 25 ± 1 °C (Peterman et al, 2003). The experiments were conducted by mechanically unzipping the dsDNA at a slow velocity of 50 nm/s to probe the potential interactions at the fork.

Single-molecule DNA unzipping data were taken at 5 kHz and later filtered to 50 Hz. The acquired data signals were converted into force and DNA extension as described previously (Zhang et al, 2019). The elasticity parameters of both the dsDNA and ssDNA for data conversion were obtained from DNA force-extension measurements in previous studies (Zhang et al, 2019). In the unzipping experiments, one separated base pair generated two nucleotides of ssDNA. Accordingly, the real-time DNA extension in nm was further converted into the number of base pairs unwound based on the elastic parameters of ssDNA under our experimental conditions. To improve the positional precision and accuracy, the force-versus-base-pairs-unzipped curves were aligned to the theoretical curve by the cross-correlation of a region before and after the ternary complex disruption (Hall et al, 2009). To account for minor instrumental drift, trapping-bead size variations and DNA linker variations, the alignment allowed for a small additive shift (< 5 bp) and multiplicative linear stretch (< 2%) using algorithms similar to those previously described (Hall et al, 2009). Please note that the

position of the pre-PAM interaction may not reflect the genuine interaction site, as the protospacer DNA was unwound upon binding by SaCas9.

## Fluorescent optical tweezer assays

Fluorescent optical tweezer assays were performed using C-trap microscopy from LUMICKS (Amsterdam, Netherlands), in which the optical tweezers were integrated with confocal microscopy and microfluidics. Wild-type Cas9 or dSaCas9 was first complexed at a 1 μM concentration with a 1:1 ratio of protein to preannealed crRNA:tracrRNA at room temperature for 10 min and subsequently diluted to 10 nM with imaging buffer (50 mM Tris–HCl pH 8.0, 100 mM NaCl, 10 mM MgCl$_2$, 100 nM TO-PRO-3 and the oxygen scavenger system with 0.8% (m/v) glucose oxidase, 2 units/μl β-D-glucose, 200 units/μl catalase and 4 mM Trolox). Bacteriophage λ DNA was labeled at either end with biotin as previously described (Gross *et al*, 2010) and was attached to 4.42 μm streptavidin-coated polystyrene particles (SPHERO) at 0.005% w/v using the laminar flow cell. For confocal imaging, two excitation wavelengths of 532 and 638 nm were used for Cy3 and TO-PRO-3, respectively. Single-molecule fluorescence signals were analyzed using software provided by LUMICKS and Zeiss.

## Stopped-flow experiments

Stopped-flow experiments were performed using an SMF400 stopped-flow instrument (Biologic). 2-AP labeled oligos were purchased from Sangon (Table EV1). dSaCas9 (1.28 μM) was assembled with sgRNA (0.64 μM) for 2 h at 37 °C in reaction buffer (50 mM Tris–HCl pH 8.0, 100 mM NaCl, and 10 mM MgCl$_2$). For each experiment, 500 μl of the 160 nM DNA template was rapidly mixed with 500 μl of 1.28 μM dSaCas9 at 37 °C in reaction buffer. The samples were excited at 310 nm, and the time-dependent fluorescence changes were monitored at 376 nm using a single-band pass filter with a 30-nm bandwidth. All experiments were repeated at least seven times for each condition, and the averaged data are shown.

## Bulk DNA cleavage and DNA footprinting assays

For the bulk DNA cleavage assays, wild-type SaCas9 was first complexed at a concentration of 0.4 μM with a 1:1 ratio of protein to sgRNA at room temperature for 10 min in reaction buffer (50 mM Tris–HCl pH 8.0, 100 mM NaCl, and 10 mM MgCl$_2$). Complexed SaCas9 (300 nM) was incubated with annealed DNA (5 nM) for 0, 15, 30, 45, and 60 min at 37 °C. The reaction was stopped by adding loading dye containing 96% formamide and 40 mM EDTA, followed by heating to 95 °C for 10 min. The reactions were analyzed by 12% denaturing polyacrylamide gel electrophoresis and phosphorimaging. Experiments were performed in triplicate and representative gels are shown.

For the DNA footprinting assays, dSaCas9 at 0.4 μM was first complexed with sgRNA at a 1:3 ratio at room temperature for 10 min in reaction buffer. Complexed dSaCas9 (300 nM) was incubated with annealed Cy5-DNA (4 nM) for 30 min at 37 °C

(Table EV1). Then, 0.2 units of Exonuclease III were added and incubated for 0, 2, 5, and 10 min at 37 °C. The reactions were stopped by adding loading dye containing 96% formamide, and 40 mM EDTA, heating to 95 °C for 10 min, and cooling slowly. The reactions were analyzed by 12% denaturing polyacrylamide gel electrophoresis and then phosphorimaged. All experiments were performed in triplicate and representative gels are shown.

## Helicase unwinding and DNAP strand-displacement replicating assays

Helicase unwinding experiments were conducted as follows (Sun *et al*, 2011). First, the dSaCas9/sgRNA complex (30 nM) was flowed into the chamber and incubated for 10 min. Second, 25 μl of 200 nM BLM helicase in DNA unwinding buffer (25 mM Tris–HCl pH 7.5, 100 mM NaCl, 1 mM MgCl$_2$, 0.1 g/ml BSA, 2 mM ATP, and 3 mM DTT) was flowed into the chamber before data acquisition. Finally, the DNA tether was stretched until the force reached 12 pN; the force remained constant while the helicase unwound the dsDNA. The DNA length was recorded in real time. DnaB helicase unwinding assays were conducted as follows. Briefly, 25 μl of 250 nM DnaB hexamer and 680 nM DnaC (the DnaB and DnaC complex is referred to as DnaB in the text) in unwinding buffer (20 mM Tris–HCl pH 7.5, 10 mM MgAc$_2$, 100 μM EDTA, 20% glycerol, 40 μg/ml BSA, 5 mM ATP, and 5 mM DTT) was flowed into the sample chamber before data acquisition. During the experiments, hundreds of base pairs of dsDNA were first mechanically unzipped to produce ssDNA for helicase loading. The DNA length was maintained until the force dropped below a threshold, indicating DnaB unwinding of the DNA fork. Finally, a constant force of 12 pN was maintained while the helicase unwound the dsDNA. The Phi29 DNAP strand-displacement replication assays were conducted similarly (Sun *et al*, 2015). In brief, 25 μl of 60 nM Phi29 DNAP in reaction buffer (50 mM Tris–HCl pH 7.5, 10 mM MgCl$_2$, 10 mM (NH$_4$)$_2$SO$_4$, and 4 mM DTT) was used for data acquisition.

The acquired data signals were converted into force and DNA extension as described previously (Hall *et al*, 2009). For the helicase unwinding studies, one unwound base pair generated two nucleotides of ssDNA. For the DNAP strand-displacement replication studies, one separated base pair was converted to one base pair of dsDNA via DNA replication and one nucleotide of ssDNA. Accordingly, the DNA extension was converted into the number of nucleotides that were unwound or replicated.

# Data availability

All data generated or analyzed during this study are included in the manuscript and supporting files. Source data files have been provided for Figs 1D, 4B, and EV4A. No primary datasets have been generated and deposited.

**Expanded View** for this article is available online.

## Acknowledgements
We thank all the staff of the molecular and cell biology core facility of the school of life science and technology at ShanghaiTech University for their

technical support. This work was supported by the National Key R&D Program of China (2016YFA0500902 and 2017YFA0106700), the Natural Science Foundation of Shanghai (19ZR1434100) and ShanghaiTech University Startup funding.

## Author contributions

BSu designed the experiments. SZ and QZ performed the experiments and analyzed the data. X-MH, H-HL, and X-GX purified the BLM and DnaB helicases and performed the stopped-flow experiments. SZ, QZ, LG, LB, XZ, and FWe designed and prepared the DNA templates for the single-molecule assays. FWa, XH, and BSh purified all Cas9 proteins. All authors contributed to the data analysis and interpretation. BSu, QZ, and SZ wrote the manuscript with input from all authors.

## Conflict of interest

The authors declare that they have no conflict of interest.

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
