## [Review Process File · EMBO Reports]

Dynamics of Staphylococcus Aureus Cas9 in DNA target Association and Dissociation

Siqi Zhang, Qian Zhang, Ximiao Hou, Lijuan Guo, Fangzhu Wang, Lulu Bi, Xia Zhang, Hai-Hong Li, Fengcai Wen, Xu-Guang Xi, Xingxu Huang, Bin Shen, and Bo Sun

DOI: [10.15252/embr.202050184](https://doi.org/10.15252/embr.202050184)

Corresponding author(s): Bo Sun (sunbo@shanghaitech.edu.cn)

Review Timeline:

Submission Date:	8th Feb 20
Editorial Decision:	12th Mar 20
Revision Received:	3rd Jun 20
Editorial Decision:	7th Jul 20
Revision Received:	11th Jul 20
Accepted:	21st Jul 20

Editor: Esther Schnapp

Transaction Report: This manuscript was transferred to EMBO reports following peer review at The EMBO Journal.

Dear Prof. Sun,

Thank you for the transfer of your manuscript together with your proposed point-by-point response to EMBO reports. I have now heard back from the referee who assessed your response, and I am happy to tell you that a revised manuscript, along the lines you suggest, would indeed be a good fit for our journal.

I would therefore like to invite you to revise your study and address all referee concerns as you propose. Please address all concerns in a complete point-by-point response. Acceptance of the manuscript will depend on a positive outcome of a second round of review. It is EMBO reports policy to allow a single round of major revision only and acceptance or rejection of the manuscript will therefore depend on the completeness of your responses included in the next, final version of the manuscript.

Revised manuscripts should be submitted within three months of a request for revision; they will otherwise be treated as new submissions. Please contact us if a 3-months time frame is not sufficient for the revisions so that we can discuss this further. You can either publish the study as a short report or as a full article. For short reports, the revised manuscript should not exceed 27,000 characters (including spaces but excluding materials & methods and references) and 5 main plus 5 expanded view figures. The results and discussion sections must further be combined, which will help to shorten the manuscript text by eliminating some redundancy that is inevitable when discussing the same experiments twice. For a normal article there are no length limitations, but it should have more than 5 main figures and the results and discussion sections must be separate. In both cases, the entire materials and methods must be included in the main manuscript file.

Regarding data quantification, please specify the number "n" for how many independent experiments were performed, the bars and error bars (e.g. SEM, SD) and the test used to calculate p-values in the respective figure legends. This information must be provided in the figure legends. Please also include scale bars in all microscopy images.

- 1) A data availability section providing access to data deposited in public databases is missing. If you have not deposited any data, please add a sentence to the data availability section that explains that.
- 2) Your manuscript contains statistics and error bars based on $n=2$ or on technical replicates. Please use scatter blots in these cases. No statistics can be calculated if $n=2$.

2) individual production quality figure files as .eps, .tif, .jpg (one file per figure).

See https://wol-prod-cdn.literatemonline.com/pb-assets/embo-site/EMBOPress_Figure_Guidelines_061115-1561436025777.pdf for more info on how to prepare your figures.

3) We replaced Supplementary Information with Expanded View (EV) Figures and Tables that are collapsible/expandable online. A maximum of 5 EV Figures can be typeset. EV Figures should be cited as 'Figure EV1, Figure EV2' etc... in the text and their respective legends should be included in the main text after the legends of regular figures.

5) a complete author checklist, which you can download from our author guidelines <<https://www.embopress.org/page/journal/14693178/authorguide>>. Please insert information in the checklist that is also reflected in the manuscript. The completed author checklist will also be part of the RPF.

6) Please note that all corresponding authors are required to supply an ORCID ID for their name upon submission of a revised manuscript (<<https://orcid.org/>>). Please find instructions on how to link your ORCID ID to your account in our manuscript tracking system in our Author guidelines <<https://www.embopress.org/page/journal/14693178/authorguide#authorshipguidelines>>

7) Before submitting your revision, primary datasets produced in this study need to be deposited in an appropriate public database (see <https://www.embopress.org/page/journal/14693178/authorguide#datadeposition>). Please remember to provide a reviewer password if the datasets are not yet public. The accession numbers and database should be listed in a formal "Data Availability" section placed after Materials & Method (see also <https://www.embopress.org/page/journal/14693178/authorguide#datadeposition>). Please note that the Data Availability Section is restricted to new primary data that are part of this study. *

Note - All links should resolve to a page where the data can be accessed. *
If your study has not produced novel datasets, please mention this fact in the Data Availability Section.

8) We would also encourage you to include the source data for figure panels that show essential data. Numerical data should be provided as individual .xls or .csv files (including a tab describing the data). For blots or microscopy, uncropped images should be submitted (using a zip archive if multiple images need to be supplied for one panel). Additional information on source data and instruction on how to label the files are available at <<https://www.embopress.org/page/journal/14693178/authorguide#sourcedata>>.

9) Our journal also encourages inclusion of *data citations in the reference list* to directly cite

datasets that were re-used and obtained from public databases. Data citations in the article text are distinct from normal bibliographical citations and should directly link to the database records from which the data can be accessed. In the main text, data citations are formatted as follows: "Data ref: Smith et al, 2001" or "Data ref: NCBI Sequence Read Archive PRJNA342805, 2017". In the Reference list, data citations must be labeled with "[DATASET]". A data reference must provide the database name, accession number/identifiers and a resolvable link to the landing page from which the data can be accessed at the end of the reference. Further instructions are available at <https://www.embopress.org/page/journal/14693178/authorguide#referencesformat>

I look forward to seeing a revised version of your manuscript when it is ready. Please let me know if you have questions or comments regarding the revision.

Kind regards,
Esther

Referee #1:

Zhang et al. investigate in this manuscript the association of SaCas9 with target DNA. This study is a follow-up of a previous study of the authors on SpCas9. The authors use mainly a DNA unzipping assay to localize spots on the DNA which are clamped by Cas9. In agreement with SpCas9 they find a spot within the target at about 8 bp downstream from the PAM and a spot about 10 bp upstream from the PAM. A difference to SpCas9 is that disruption of the upstream interaction does not necessarily disrupt the downstream interactions. The authors investigate also a set of shortened DNA targets with a limited number of matching bases proximal to the PAM, where they use the unzipping assay to probe binding, a FRET assay to probe DNA unwinding in a certain region and bulk cleavage assays. They furthermore test whether different molecular motors can displace Cas9 from the DNA and find that SaCas9 is much more resistant against motor activity than SpCas9. Overall this is an extensive and careful study about the stability of SaCas9 bound to target DNA. The study corroborates, however, largely previous results on SpCas9. Furthermore, rather limited new mechanistic insight on Cas9 is provided. It is not clear to me how relevant these unzipping assays are that seem to probe only very strong interactions but not weaker interactions that are important for the regulation of the Cas9 complex, such as the engagement of the HNH domain.

Minor comments:

- 1) I don't think the authors can provide realistic K_d measurements, since the dissociation time of Cas9 is extremely large (several hours). Either the authors show kinetic measurements and demonstrate that the association reaction has really equilibrated or remove the K_d measurements.
- 2) The occurrence of two peaks for the one trace in figure 1C is hard to see, since it is overlaid by the single peak trace. A different visualization should be found for this.
- 3) The authors should be more precise throughout when speaking about the possibility of SaCas9 to unwind DNA, since they probe unwinding only in a very limited region.
- 4) The English should be improved throughout. Some parts are really hard read by gluing too many terms together (e.g. "We find that its stable DNA binding and unwinding necessitate the complementarities of 6- and 14-bp PAM-proximal DNA with guide RNA, respectively.", "PAM-proximal RNA-DNA heterduplex matching number")
- 5) Revisit: "The type II CRISPR-Cas system consists of a single Cas9 protein...". There are several Type II systems and they contain also other proteins. Just their effector complex (cas9) is a single protein and not a multisubunit complex.

Referee #2:

Zhang et al present their single-molecule work on SaCas9 DNA interactions. They use various molecular biology and biophysics tools including optical tweezers, single molecule FRET, DNase foot printing assays to study the mechanistic aspects of the interactions between SaCas9/crRNA/target DNA. Their claims are: (1) there are two major interaction sites when probed by unzipping of the target DNA, (2) dSaCas9 binds strongly to the target DNA such that it cannot be displaced by most of DNA processing motors, (3) 6-bp matching

is required for binding of SaCas9, while 18-bp matching is needed for efficient cleavage, (4) SaCas9 releases PAM-distal DNA after cleavage, but not the PAM-proximal DNA, and (5) after cleavage, the post-PAM interaction changes and may be broken by some DNA processing motors. The experiments were well designed and appropriate to address the questions they raised. However, this reviewer finds this comparative study lacks novelty and is missing some crucial data and would recommend publishing it only after major revision.

Major comments

1. They find that saCas9 remains bound to one of the cleaved DNA (Fig. 4), while they explain that SaCas9 is an effective multi-turnover enzyme (page 20, last sentence). How is it possible that an enzyme that does not release all the product after reaction can be an effective multi-turnover enzyme? This reviewer finds the explanation by the authors not sufficient.
2. On page 7, the authors mentioned that "..., they continued to be similar to those of the corresponding naked DNA, ...": I do not see any similarity to the naked DNA behavior after rise of the force in Fig. 1B. Instead, I see a gradual decrease in force, which has not been seen at all with the naked DNA. There is no explanation on this gradual decrease in the text. I can only assume that their speed of unzipping (i.e. increase in DNA length) was too fast so that the remaining dsDNA beyond the disruption site gradually unzipped before the trapping force was relaxed after disruption. If so, they need to provide evidence that their unzipping speed is optimal and their observation and conclusion is still valid regardless of the unzipping speed they used.
3. On a related note to the above issue, the apparent disruption forces measured in Fig. 1D, 3B, S5, S8, S11 can be different when the unzipping speed changes. The authors should explain how much different force would be measured when a speed different from 50nm/sec (a standard speed the authors used) is used. Also it would be helpful for the broad audiences if they explain the measured disruption forces and the unzipping time scale are in the biologically relevant range regarding the Cas9-mediated reactions.
4. The disruption force of the post-PAM interaction in Fig 1D is ~30pN, whereas that in Fig. S11 is ~50pN. A simple-minded conclusion from these two observations is that the post-PAM gets stronger after cleavage. This is clearly contradicting with their conclusion with the BLM-induced displacement in Fig. 4F, where they find weakened saCas9. However, in the Discussion and Results, the authors fail to provide a clear explanation on this contradiction.
5. On page 8 the authors talk about an "unexpected interaction site at approximately 6bp downstream of the PAM", which is outside the apparent footprint of the protein (39bp), however no direct proof of this being a genuine interaction rather than a steric effect due to the flexibility and movement of the protein domains is provided. As the authors mention, indeed the crystal structure does not show any interactions at that point thus making it unlikely that any amino acid residues of the protein are directly interacting with the DNA 6-10 bp downstream of the PAM.
6. On page 12 the authors talk about "traces with only four matches showing the absence of stable binding." However they do not show such traces. Similarly, they talk about traces showing stable binding, but such traces are never shown.
7. In Figure S7 the authors show the percentage of molecules bound vs the mismatch extent. It seems that an abrupt major transition happens between 4-6 nt, however data for the middle point (5nt) at which the transition may be happening and give more insight into the behaviour of SaCas9 are missing.
8. In Figure 3C the authors should show the full FRET trace including the donor and acceptor signals, as well as a reference trace showing the FRET behaviour of naked DNA.
9. There is no reporter dye on any part of the SaCas9:sgRNA complex which would make it possible to observe binding. Consider placing a dye on the sgRNA instead of the target strand

to monitor the arrival of Cas9 as well as unzipping behaviour.

10. Several other smFRET papers on SpCas9 binding have shown fluctuations upon binding and unzipping of the DNA duplex (doi:10.1038/ncomms13350), <https://doi.org/10.1021/jacs.8b03102>. Since the core folds of both SaCas9 and SpCas9 are quite similar, it is surprising that the authors did not observe any dynamic behaviour, either for partially mismatched or full targets, even though the FRET histograms in figure 3D for partially mismatched targets are ambiguous and can contain more than 2 FRET peaks. Please comment.

Minor comments

1. The authors use somewhat strange English expressions. The text is somehow understandable, but it does not read smoothly at all. It would be greatly beneficial if the manuscript can be revised by a native English speaker with expertise in biology. Correct typos and consider heavily copy-editing the text as some words are missing and some sentences are difficult to follow.
2. The language used in the manuscript is confusing: the use of pre-PAM and post-PAM interactions indicate interactions that happen prior to PAM binding and following PAM recognition as pre- and post- usually refer to time, rather than position. Consider replacing them with upstream/downstream.
3. The authors' claim that they show results with "near base-pair resolution" is too strong for the data they present as an uncertainty of 4-10 bp in some of their measurements is 25%-50% of the total DNA length that is of interest (length of the target).
4. In Introduction it is mentioned that sgRNA comprises crRNA and tracrRNA which is factually incorrect as sgRNA does not contain crRNA and tracrRNA but is a chimera of the two.
5. Fig. 3A: please indicate how many molecules were tested to calculate the percentages.
6. In figure 3B the error bars are so large that essentially there is no real difference between any of the constructs
7. Fig. 3D: 3D presentation of multiple graphs looks fancy but can be somewhat deceiving because it is very difficult to check the shape of the histogram. Also a quantitative comparison between the peaks from the graph is not possible. It is always a better practice to present a data set in a way that the values can be directly readable from the graph.
8. Consider adding a supplementary figure that would show the Gaussian fits of the histograms in Figure 3D for clarity.
9. Fig. 4C right: Why is there only one bead present with WT SaCas9? Wasn't it a dual trap optical tweezers?
10. Fig. S12: I can only assume that the data in Fig. S12 is measured without SaCas9. Is it correct? Also, in the legend, they mentioned about Cy5 signal. Wasn't it Cy3 they labeled the crRNA? Also, there is a typo (crDNA>crRNA)
11. Fig. S2: Please specify how many DNA molecules they measured.
12. On page 3 the authors mention that SaCas9 and SpCas9 share 17% sequence similarity, but do not mention that despite the low sequence similarity the two proteins share very similar core folds.
13. On page 3, it is written that „ensuing the complementarity of the 20 bp RNA-DNA hybrid, SpCas9 undergoes a significant structural rearrangement to adopt a bilobed architecture ". This is incorrect, SpCas9 adopts a bilobed architecture after binding sgRNA. The global structural rearrangements after binding the target DNA are minimal (not counting the domain movements required for cleavage).

Referee #3:

In this work, Zhang & Zhang et al. probe energetic features of the RNA-guided interactions of SaCas9 with a DNA target using a series of optical tweezers experiments, complemented by biochemical assays that provide information about enzyme catalytic efficiency and FRET assays claimed to provide information about DNA unwinding. Having already applied many of these techniques to SpCas9 in (Zhang et al., *Science Advances*, 2019), this group has implemented the same experimental workflow on SaCas9 in the present study, allowing comparison of the two Cas9 orthologs. In the context of growing interest in the biotechnological use of SaCas9, the data presented in this work provide a catalog of biophysical parameters for SaCas9 that were previously only known for SpCas9. The authors demonstrate a few differences between the two orthologs that may have biotechnological significance, which include:

- Catalytically inactive SaCas9 (dSaCas9) cannot be dissociated from DNA targets by processive molecular motors that were able to dissociate dSpCas9 under equivalent conditions. This result suggests that dSaCas9 may remain bound to DNA longer than dSpCas9 *in vivo*.
- WT SaCas9 releases the PAM-distal DNA fragment following cleavage, whereas SpCas9 remains bound to both product fragments. Given the recently reported multiple-turnover DNA cleavage activity of SaCas9 (compare SpCas9, which conducts single-turnover DNA cleavage), this result suggests that substrate cycling may begin with fast dissociation of the PAM-distal fragment.

Major concerns:

1. The authors use a FRET assay similar to that described in Singh et al. (2018) NSMB to make the claim that SaCas9 requires 14 PAM-proximal matches between the crRNA and the DNA target to achieve DNA "unwinding." There are problems with the authors' experimental design and their interpretation of the data. I will first explain the informative experiments performed by Singh et al., and then I will explain why the analogous experiments performed by Zhang & Zhang et al. are not informative.

Singh et al. performed their experiments using constructs with both dyes conjugated within the target sequence—this experimental setup was designed to detect the presence of an intermediate unwinding state hypothesized to have somewhere between 8 and 16 PAM-proximal DNA base pairs unwound. Singh et al. then measured differences in the relative population of the intermediate/fully-unwound states on high-fidelity SpCas9 mutants bound to DNA targets with up to 4 PAM-distal mismatches. They found that, in the presence of PAM-distal mismatches, the intermediate unwinding state was more highly populated on the high-fidelity mutants than on the wild-type protein, suggesting a conformational basis for slowed cleavage of mismatched DNA targets by high-fidelity Cas9 mutants.

Zhang & Zhang created a DNA construct that contained a Cy3-dT at position 13 of the target strand and a Cy5-dT at position 24. This construct would not be expected to yield a change in FRET unless unwinding proceeded past position 13 of the target sequence. Therefore, without even looking at the data, it is confusing why the authors conducted the experiments with the 5-22mm, 7-22mm, 9-22mm, and 11-22mm constructs (which do not have enough RNA complementarity to allow DNA unwinding past position 13) in the first place.

Additionally, it is difficult to explain why the 7-22mm, 9-22mm, and 11-22mm experiments do in fact contain significant low-FRET populations. It is possible that in these populations, the dyes are pressed against the protein and consequently are restricted in their spatial and/or orientational freedom to a low-FRET ensemble. Therefore, the only experiment with an interpretable result is the comparison of the "Full match" construct to the "15-22mm"

construct. With this comparison alone, it is difficult to ascertain whether the FRET distribution in the 15-22nm experiment represents a meaningful partition between two stable unwound states or a product of the same assay artifact that yielded the results for the constructs containing more mismatches. Furthermore, even in a best-case scenario where the authors were able to definitively identify a stable intermediate unwinding state, the scientific value of this finding would rely on extensive additional experimentation linking the result to propensity for off-target cleavage (which was the main point of Singh et al. 2018), which is not relevant to the remainder of the present manuscript. Why did the authors choose to monitor this particular reaction coordinate, and what does it tell us about the mechanism of SaCas9? In the absence of a strong answer to these questions, I would advise omitting the FRET data and corresponding conclusions from the manuscript.

2. The remainder of the data support the main conclusions of the paper, and I believe they can stand alone in the absence of the FRET data. However, the main text of the paper is rife with smaller conclusions or explanations that are presented with imprecise language and/or an unclear experimental basis, due in some cases to poor writing quality and in others to overinterpretation (some examples are presented in "minor concerns"). These problems are so widespread that I consider the text, collectively, to be a major concern. Extensive care should be taken to improve the clarity and precision of the writing throughout the manuscript.

Minor concerns:

1. The interpretations of the unzipping experiments, which are the cornerstone of this paper, require more careful consideration of the physical nature of the events that occur when the unzipping fork reaches the Cas9 complex. The authors invoke a conceptual framework described in Koch et al., *Biophys J*, 2002, in which proteins bound to double-helical DNA substrates undergo cooperative dissociation events in response to force-induced DNA base-pair disruption. The described experimental setup, when applied to Cas9, requires a completely different mode of interpretation due to Cas9's R-loop-mediated interaction with DNA. Within the complex, the DNA is not even base paired, eliminating base-pair disruption as a relaxation pathway. Instead, given the relatively loose association of the non-target strand with the protein in known Cas9 structures, the system probably relaxes (in response to applied force) through breakage of interactions between the non-target strand and the protein (within the R-loop) and/or the target strand (within the PAM). It may even be the case that the non-target strand is torn out of the complex without disrupting Cas9's crRNA-mediated association with the target strand, as it is known that SaCas9 can stably bind to a single-stranded target strand (this possibility invalidates the statement "...indicating that the ternary complex was completely disrupted"). I encourage the authors to consult "Direct Observation of the Formation of CRISPR-Cas12a R-loop Complex at the Single-Molecule Level," Cui et al., *ChemComm*, 2020, for an explanation of the likely relaxation pathway for a similar system. The authors should adjust their writing throughout the manuscript to explain that their data cannot be interpreted in the same way as data for proteins that bind double-helical DNA.

2. The interpretation problem described in Comment #1 also poses a problem for the presentation of data in which "position" is expressed in terms of base pairs (ex. Fig. 1B). It is physically meaningless to describe a force peak's "position" within the R-loop in terms of base pairs, as base pairs are not being broken. Given the complex and unknown nature of the relaxation pathway, it may even be physically meaningless to ascribe any sort of "position" to a force peak at all (relaxation may involve simultaneous breakage of interactions at multiple locations within the complex). If one assumed that the relaxation pathway involved sequential breakage of non-target-strand:protein interactions along an axis parallel to that of

the RNA:DNA hybrid, one could adjust the model to incorporate the compressed length of the A-form helix in one of the arms leading away from the fork-this would imply that the so-called "pre-PAM" interaction is actually closer to the PAM than in the authors' model (and may even be the protein's interaction with the PAM itself). However, with so little known about the relaxation pathway, the authors could simply acknowledge that they are unable to assign a precise spatial "position" to any force peak in the forward unzipping experiments or to intra-R-loop force peaks in the reverse unzipping experiments (note that this uncertainty does not apply to the first force peak of reverse unzipping experiments, which is approached from an unperturbed DNA duplex). This may involve changing the x-axis units of their unzipping plots to a measure of extension length (or a similar model-independent physical distance that is defined by the displacement of the optical tweezers), accompanied by diagrams that indicate the hypothesized "position" of the interaction only in very general terms (ex. "somewhere within the R-loop" or "near the PAM"). The text should also be adjusted accordingly. For example, I find the following sentences unacceptable because they reveal a failure to correctly interpret what can be known about a given interaction based on the extension length at its associated force peak: "The other pre-PAM interaction between dSaCas9 and DNA is located within the DNA-RNA heteroduplex region." "Even though the structural data showed that the C-terminal region of the REC lobe of SaCas9 interacts with the PAM-distal region of the heteroduplex, ..." (Objection: It is unknown whether breakage of a heteroduplex:protein interaction accounts for relaxation of this force peak, and I would argue that it is in fact more likely that we are observing breakage of a non-target-strand:protein interaction. The authors should remain agnostic.)

3. In Fig. 1E, the gel should not be cropped so tightly at the bottom. The reader should be able to ascertain that the exonuclease did not proceed more deeply into the complex.

4. In Fig. 2C/D, BLM is depicted to proceed in two different directions. What is the correct polarity of its processive unwinding activity?

5. In Fig. 3F, it is misleading to present single-timepoint cleaved fractions, which reflect an arbitrarily chosen timepoint, on the same scale as the "Binding" and "Unwinding" parameters, which reflect a meaningful equilibrium (ex. the cleavage curve may have looked identical to the binding curve if the authors had chosen a later timepoint). The "Cleavage" data should be depicted in a separate plot or on its own axis that indicates "Fraction cleaved at X timepoint".

6. When I zoom in on the background of the micrographs in Fig. 4C, the Cy3 puncta in the background are much brighter than those in the background of Fig. S12, even though the two should be equivalent based on my understanding of the methods. The authors should make sure they are collecting and presenting their micrographs equivalently.

7. How did the authors verify the following statement? "In addition, the bound dSaCas9's lifetime was longer than 2 hours."

8. In Fig. S4, how many reverse unzipping events were conducted at each Cas9 concentration? This should be indicated in the figure legend.

9. The first sentence of the intro contains an incorrect statement. "Cas" stands for "CRISPR-associated" (not "CRISPR-associated system") and refers to the protein-coding genes of CRISPR-Cas systems.

10. The second sentence of the intro contains an incorrect statement. The entity described by the authors is the type II interference complex, not the type II CRISPR-Cas system.

11. The last sentence on page 3 contains an incorrect statement. The bilobed architecture of Cas9 is an intrinsic feature of the protein in all known ligand states-it is not only adopted after DNA-induced rearrangement.

12. The clause "even though the post-PAM interaction between SaCas9 and DNA becomes stronger after cleavage (Supp. Fig. 11)" is an overinterpretation. The increased interaction

strength shown in Supp. Fig. 11 could just as well be due to a direct stabilizing effect of Mg^{2+} (and not the release of the PAM-distal fragment). The authors would need to do a Mg^{2+} -free experiment with an already-cleaved DNA substrate to verify their claim.

Non-essential suggestions for improving the study:

1. In Fig. 4C, the Cy3 spot on the end of the DNA is barely visible and not so different from other background Cy3 spots. The authors should consider an additional (or alternative) mode of data presentation/analysis that can assure the reader that there is indeed a stable spot there.
2. While the terms "downstream/upstream of the PAM" are useful when defined in a figure, their arbitrary definition makes them confusing when reading text alone and should be avoided as much as possible, especially in the abstract.
3. The authors have questionable word choice in several places (examples: ensuing, testify, intense, formidable, asks for).
4. The authors should include a written description of the physical process occurring during the slow linear drop in force following each "dissociation" event. After reading the sentence : "after the rise in force, they continued to be similar to those of the corresponding naked DNA," I was initially confused as to why the protein-containing traces did not resemble the gray traces.
5. The text in the section titled "dSaCas9 presents a strong barrier to DNA tracking motors" could be significantly condensed because the experimental setup was the same for all motors, and the results were nearly equivalent for all motors.

Response to Referees

Manuscript #	EMBOR-2020-50184-T
Title	Dynamics of Staphylococcus Aureus Cas9 in DNA target Association and Dissociation
Corresponding Author	Bo Sun (ShanghaiTech)
Contributing Authors	Siqi Zhang, Qian Zhang, Fengcai Wen, Lijuan Guo, Lulu Bi, Xia Zhang, Fangzhu Wang, Xi-Miao Hou, Xu-Guang Xi, Xingxu Huang, Bin Shen

We greatly appreciate the valuable comments and suggestions from the referees and the opportunity to improve the manuscript. We have considered each of the comments and have carefully addressed them, as detailed in the following responses. The changes made to the manuscript are highlighted in red in the main text. We begin below with a summary of the major changes. This is followed by a detailed, point-by-point response to each comment. Each comment below is shown in black (bold) and is followed by our response in blue (not bold).

Summary of major changes to the manuscript:

1. To precisely detect protospacer DNA unwinding by SaCas9, we have conducted a stopped-flow assay with a DNA substrate containing 2-aminopurine (2-AP) at position +18 of the non-target strand to report the late stage of the unwinding. This method is more accurate in detecting DNA unwinding compared to FRET, as base-pairing changes are reflected by a single sensitive probe. Moreover, it also allowed us to measure the unwinding (R-loop formation) rate. We found that 18 RNA-DNA matches are required for SaCas9 to form a stable R-loop which occurred within one second. This set of data has now been added as the new Fig 4C and D, Fig 5 has been updated accordingly and the FRET data have been removed, as suggested by referee #3.
2. We have added data from the DNA unzipping experiments to illustrate that the lifetime of DNA-bound dSaCas9 is over 24 hours, and the K_d measurements for dSaCas9 have been removed, as the equilibrium state could not be reached within a reasonable time. We have also demonstrated that WT SaCas9 can completely dissociate from the DNA within hours after cleavage, providing direct evidence for SaCas9 being a multiple-turnover enzyme. These data have now been added as the new Figs EV2G and EV5C.
3. As suggested by referee #2, we have performed DNA unzipping experiments with 5-nt matched sgRNA and did not detect DNA binding of dSaCas9, corroborating our conclusion that at least 6 bp matches are required for the stable DNA binding of dSaCas9. This set of data has now been added as the new Figs 3A, EV4A and B.

4. We have repeated the DNA cleavage assay with more time points and have updated the gel picture in the new Fig 3E. The old Fig 3F showing the fractions bound, unwound and cleaved as a function of mismatches has been removed. We have also updated the DNA footprinting gel picture, showing no other cleavage products beside the single band (new Fig 1E).
5. As suggested by referee #3, we have conducted DNA unzipping assays with wild-type SaCas9 in the absence of Mg^{2+} . Without Mg^{2+} , the unzipping signatures with SaCas9 resemble those with dSaCas9 in terms of both positions and strengths and the post-PAM interaction is indeed relatively weaker than that with Mg^{2+} . These results suggest that the stronger post-PAM interaction detected with wild-type SaCas9 in the presence of Mg^{2+} possibly results from a stabilizing effect of Mg^{2+} , as suggested by the referee. This set of data has now been added as the new Fig EV5A.
6. We have also made a few other changes to the manuscript, as suggested by the referees, and this manuscript has been edited by an English language editor. These changes are indicated in our responses to the comments from the referees.

Referee #1:

Zhang et al. investigate in this manuscript the association of SaCas9 with target DNA. This study is a follow-up of a previous study of the authors on SpCas9. The authors use mainly a DNA unzipping assay to localize spots on the DNA which are clamped by Cas9. In agreement with SpCas9 they find a spot within the target at about 8 bp downstream from the PAM and a spot about 10 bp upstream from the PAM. A difference to SpCas9 is that disruption of the upstream interaction does not necessarily disrupt the downstream interactions. The authors investigate also a set of shortened DNA targets with a limited number of matching bases proximal to the PAM, where they use the unzipping assay to probe binding, a FRET assay to probe DNA unwinding in a certain region and bulk cleavage assays. They furthermore test whether different molecular motors can displace Cas9 from the DNA and find that SaCas9 is much more resistant against motor activity than SpCas9. Overall this is an extensive and careful study about the stability of SaCas9 bound to target DNA. The study corroborates, however, largely previous results on SpCas9. Furthermore, rather limited new mechanistic insight on Cas9 is provided. It is not clear to me how relevant these unzipping assays are that seem to probe only very strong interactions but not weaker interactions that are important for the regulation of the Cas9 complex, such as the engagement of the HNH domain.

Response: We thank the referee for his/her positive remarks and would like to share our thoughts on the novelty of this work.

First, a detailed mechanistic study of SaCas9 is necessary and in high demand. CRISPR-Cas systems have been widely repurposed for many biological and medical usages. Mechanistic studies of the major proteins in these systems would not only provide instruction for their better usage as genome tools but also aid in the expansion of their applications. One of the reasons that SpCas9 is commonly used for biological and medical applications is that it has been extensively studied, and its molecular mechanisms have been largely revealed. However, contrary to the case for SpCas9, the molecular mechanisms of SaCas9 have remained largely unknown, limiting its effective modifications and applications. In addition, different Cas proteins may possess distinguished mechanistic features that are suitable for different purposes. For example, the object of this study, SaCas9, is a relatively small enzyme that facilitates *in vivo* delivery for therapeutic applications. The distinguishing features of SaCas9 may make it a better tool for some applications compared to SpCas9. Our work here is indeed filling this knowledge gap.

Second, in this work, we aim to provide a complete dynamic picture of SaCas9, displaying its interplay with DNA from association to dissociation. Using a series of single-molecule and ensemble assays, we examined the DNA binding, unwinding, cleavage, and dissociation of SaCas9. Moreover, a detailed mechanistic comparison between SaCas9 and SpCas9 helps summarize the general characteristics of Cas9 proteins and their distinguishing mechanistic features. Our previous work (DOI: 10.1126/sciadv.aaw9807) to which the referee referred, however, mainly focused on the identification of the post-PAM interaction of SpCas9 with DNA and its regulation of SpCas9 activity. The current work can be considered more than a follow-up of our previous study.

Third, novel mechanistic features of Cas9 proteins have been provided in our study. As pointed out by both referees #2 and #3, this work revealed the mechanistic differences between SaCas9 and SpCas9 in a few aspects. 1) Compared with dSpCas9, dSaCas9 has a stronger binding affinity to the target DNA, and DNA repair helicase cannot remove DNA-bound dSaCas9. This finding provokes *in vivo* studies examining dSaCas9 as a more effective obstacle on DNA compared to dSpCas9. It is noteworthy that the characteristics of DNA binding of Cas9 proteins that we provided in this work are one of the considerations when using them to perturb DNA-based transactions, such as transcription and replication. 2) Compared with SpCas9, the SaCas9 protein is more tolerant to off-target sites, as only six PAM-proximal matches are required for its stable DNA binding, though matches at positions 1-2 are indispensable. These findings provide information for sgRNA design in future *in vivo* applications. 3) Unlike SpCas9, which stably binds to both DNA ends after cleavage, we showed that SaCas9 autonomously releases both of the cleaved DNA ends. This finding explains why SaCas9 can act as a multi-turnover enzyme. This is also a new finding for Cas proteins.

We agree with the referee that the relatively transient and weak interactions between SaCas9 and DNA might be too subtle to be detected in our DNA unzipping assays. However, we believe that the stable and strong interactions that we identified in this work also provide valuable insights in understanding the molecular mechanisms. First, we directly quantified the interactions sites at a high resolution, along with their corresponding strengths. The structural study cannot offer information on the strengths of the interaction sites directly related to the stability of SaCas9 on DNA and to its disassociation. As we showed in this work, the fact that DNA-bound dSaCas9 is more resistant to motor proteins compared to dSpCas9 is because of its much stronger pre-PAM interactions with DNA. Modifications of these interaction sites may be carried out to adjust its dissociation rate and further increase its efficiency as a genome tool. Second, an unexpected post-PAM interaction site beyond the PAM and the protospacer has been identified. A post-PAM interaction has also been detected for SpCas9, as reported in our previous work, albeit at a different position (DOI: 10.1126/sciadv.aaw9807). In addition, a similar interaction site has also been observed with Cpf1 (data not shown). It is reasonable to speculate that this post-PAM interaction site is a conserved feature of these proteins and plays an important role in regulating their activities. This is also a new finding in the field. We are currently working on understanding the potential functions of this post-PAM interaction, and our preliminary data suggest that the interaction site is essential for R-loop formation (data not shown).

Minor comments:

1) I don't think the authors can provide realistic K_d measurements, since the dissociation time of Cas9 is extremely large (several hours). Either the authors show kinetic measurements and demonstrate that the association reaction has really equilibrated or remove the K_d measurements.

Response: We thank the referee for pointing out that we may not have reached the equilibrium state in our assays for K_d measurements. We have performed an additional set of experiments to measure the dissociation of DNA-bound dSaCas9 in the absence of free proteins. It turns out that the lifetime of DNA-bound dSaCas9 is over 24 hours (New Fig EV2G). The equilibrium state could not be reached within a reasonable time, and we thus decided to remove the K_d measurement data, as suggested.

2) The occurrence of two peaks for the one trace in figure 1C is hard to see, since it is overlaid by the single peak trace. A different visualization should be found for this.

Response: We apologize for not being clear in the data presentation. We have provided separated zoomed-in figures alongside the traces (New Fig 1B and C) to highlight the peaks of the traces, which should make the information more readable to the audience.

3) The authors should be more precise throughout when speaking about the possibility of SaCas9 to unwind DNA, since they probe unwinding only in a very limited region.

Response: We apologize for overstating our results. As mentioned above, we have replaced the FRET results with another set of data from a stopped-flow assay. In this assay, we used a DNA substrate that included a 2-AP at position +18 of the non-target strand to report on the late stage of DNA unwinding (New Fig 3C and D). As suggested by the referee, we were very careful in stating the results in the revised manuscript to avoid overstatements.

4) The English should be improved throughout. Some parts are really hard read by gluing too many terms together (e.g. "We find that its stable DNA binding and unwinding necessitate the complementarities of 6- and 14-bp PAM-proximal DNA with guide RNA, respectively.", "PAM-proximal RNA-DNA heterduplex matching number")

Response: We apologize for the inconvenience. We have consulted an English language editor to help us improve the language. We hope that the referee is now satisfied with the revised manuscript.

5) Revisit: "The type II CRISPR-Cas system consists of a single Cas9 protein...". There are several Type II systems and they contain also other proteins. Just their effector complex (cas9) is a single protein and not a multi-subunit complex.

Response: We apologize for this mistake and have corrected it in the revised manuscript.

Referee #2:

Zhang et al present their single-molecule work on SaCas9 DNA interactions. They use various molecular biology and biophysics tools including optical tweezers, single molecule FRET, DNase foot printing assays to study the mechanistic aspects of the interactions between SaCas9/crRNA/target DNA. Their claims are: (1) there are two major interaction sites when probed by unzipping of the target DNA, (2) dSaCas9 binds strongly to the target DNA such that it cannot be displaced by most of DNA processing motors, (3) 6-bp matching is required for binding of SaCas9, while 18-bp matching is needed for efficient cleavage, (4) SaCas9 releases PAM-distal DNA after cleavage, but not the PAM-proximal DNA, and (5) after cleavage, the post-PAM interaction changes and may be broken by some DNA processing motors. The experiments were well designed and appropriate to address the questions they raised. However, this reviewer finds this comparative

study lacks novelty and is missing some crucial data and would recommend publishing it only after major revision.

Response: We thank the referee for his/her positive remarks and refer the referee to our response to the first comment from referee #1 about the novelty of this work. We have addressed individual comments below.

Major comments

1. They find that saCas9 remains bound to one of the cleaved DNA (Fig. 4), while they explain that SaCas9 is an effective multi-turnover enzyme (page 20, last sentence). How is it possible that an enzyme that does not release all the product after reaction can be an effective multi-turnover enzyme? This reviewer finds the explanation by the authors not sufficient.

Response: We thank the referee for pointing out that our data were not sufficient to draw the conclusion. We agree with the referee that SaCas9 is required to release both cleaved DNA ends to become an effective multi-turnover enzyme. To examine this characteristic, we have conducted an additional DNA unzipping experiment to address whether SaCas9 in the presence of competitor DNA can release both DNA ends after cleavage. It turns out that after the release of the PAM-distal DNA, SaCas9 can also autonomously dissociate from the PAM, though it takes several hours. However, it is still more efficient compared to SpCas9. These data provide direct evidence for SaCas9 being a multiple-turnover enzyme and have been added as the new Fig EV5B and C.

2. On page 7, the authors mentioned that "..., they continued to be similar to those of the corresponding naked DNA, ...": I do not see any similarity to the naked DNA behavior after rise of the force in Fig. 1B. Instead, I see a gradual decrease in force, which has not been seen at all with the naked DNA. There is no explanation on this gradual decrease in the text. I can only assume that their speed of unzipping (i.e. increase in DNA length) was too fast so that the remaining dsDNA beyond the disruption site gradually unzipped before the trapping force was relaxed after disruption. If so, they need to provide evidence that their unzipping speed is optimal and their observation and conclusion is still valid regardless of the unzipping speed they used.

Response: We apologize for the improper data presentation and explanation. In our DNA unzipping assays, the DNA trunk to be unzipped was ~ 1,800 bp, and ~ 1,000 bp of dsDNA remained after the disruption of the DNA-bound SaCas9. After the disruption, the force dropped to ~ 17 pN (naked DNA unzipping force) in up to 300 bp, which was followed by the DNA unzipping signature in the remaining ~ 700 bp (see data below). In the previous manuscript, we meant to express that the unzipping signature of the remaining 700 bp was similar to that of naked DNA, as shown below. In the updated figure, we show complete unzipping traces and their

zoomed-in views to highlight the protein disruption force. We have also revised the manuscript by stating:

“the forces dropped to ~ 17 pN after the rise and continued to be similar to those of the corresponding naked DNA”.

When protein disruption occurs, the DNA is under extreme tension (up to 60 pN). After disruption, the force is still much higher than the typical DNA unzipping force (~ 17 pN), and it thus suddenly unzips the following dsDNA to generate new ssDNAs in less than one second. This newly generated ssDNA relaxes the DNA tether and reduces the tension on the DNA until the naked DNA unzipping force (~ 17 pN) is reached. Under our experimental conditions, the sudden tension reduction from ~ tens of pN to 17 pN requires up to ~ 300 bp of dsDNA to be unzipped. We would like to kindly remind the referee that the amount of suddenly unzipped DNA heavily relies on the disruption force instead of the unzipping speed. As the referee can tell between the two representative traces in Fig 1C [Figures for referees not shown.], the higher the disruption force is, the more dsDNA is suddenly unzipped. Although the disruption force indeed depends on the unzipping speed, the change over a wide speed range is too trivial to significantly affect the unzipping signature. We have provided an explanation for the choice of the unzipping speed in our response to the next comment. Please see our response to the next comment.

3. On a related note to the above issue, the apparent disruption forces measured in Fig. 1D, 3B, S5, S8, S11 can be different when the unzipping speed changes. The authors should explain how much different force would be measured when a speed different from 50nm/sec (a standard speed the authors used) is used. Also it would be helpful for the broad audiences if they explain the measured disruption forces and the unzipping time scale are in the biologically relevant range regarding the Cas9-mediated reactions.

Response: The referee is right, as the disruption force of the protein does depend on the unzipping speed (DOI: 10.1103/PhysRevLett.91.028103). The DNA unzipping is a thermally activated off-equilibrium process. The slower the unzipping speed is, the closer it is to the equilibrium state. The disruption force at the equilibrium state is

the smallest and reflects the genuine interaction strength. To examine whether the unzipping speed that we chose to use is reasonable, we have conducted DNA unzipping experiments under a wide range of unzipping speeds (from 10 to 500 nm/s). Please note that a few nm/s of DNA unzipping speed is believed to make the disruption close to the equilibrium state. Our data showed that the disruption forces were comparable under these unzipping speeds, meaning that the unzipping speed does not significantly affect the protein disruption force within this speed range. We have incorporated these data in Fig EV 2C and F.

The reason why we chose to use 50 nm/s of unzipping speed is as follows. The unzipping fork mimics the motion of DNA-based molecular motors, such as helicases and polymerases. These motors typically move along the DNA at a speed of tens of bp per second, corresponding to tens of nm/s under our experimental conditions. In this work, one of the aims was to examine the consequence of a collision between a molecular motor and a DNA-bound SaCas9. We thus chose to use 50 nm/s. We thank the referee for the suggestion and have incorporated this discussion into the revised manuscript on page 6.

4. The disruption force of the post-PAM interaction in Fig 1D is ~30pN, whereas that in Fig. S11 is ~50pN. A simple-minded conclusion from these two observations is that the post-PAM gets stronger after cleavage. This is clearly contradicting with their conclusion with the BLM-induced displacement in Fig. 4F, where they find weakened saCas9. However, in the Discussion and Results, the authors fail to provide a clear explanation on this contradiction.

Response: It is true that the detected disruption force of the post-PAM interaction increased after cleavage. However, dSaCas9 associates with the target DNA via more than one interaction site. In addition to the post-PAM interaction, the strong pre-PAM interaction tightly clamps the protein onto the DNA, possibly acting alongside other weak interactions in the protospacer region that could not be detected in our assays. BLM has to overcome these multiple barriers, rather than a single relatively stronger post-PAM interaction, to displace the protein. For WT SaCas9, the cleavage allows it to release the PAM-distal DNA, and the pre-PAM interaction consequently disappears. Thus, even though the post-PAM interaction becomes stronger after cleavage, the overall energy required for BLM to displace the protein reduces as the pre-PAM interaction disappears. We apologize for not being able to provide this explanation in the original manuscript and have incorporated it into the revised manuscript in the discussion by stating:

“Moreover, we demonstrated that even though the post-PAM interaction of wild-type SaCas9 is stronger than that of dSaCas9 (Figs 1D and EV5A), a downstream helicase can still displace it (Fig 4F). These findings also suggest that the DNA binding affinity of dSaCas9 is likely dominated by the pre-PAM interaction, and the disruption

of the post-PAM interaction is not enough to dissociate the dSaCas9 from its target DNA.

5. On page 8 the authors talk about an "unexpected interaction site at approximately 6bp downstream of the PAM", which is outside the apparent footprint of the protein (39bp), however no direct proof of this being a genuine interaction rather than a steric effect due to the flexibility and movement of the protein domains is provided. As the authors mention, indeed the crystal structure does not show any interactions at that point thus making it unlikely that any amino acid residues of the protein are directly interacting with the DNA 6-10 bp downstream of the PAM.

Response: We understand the referee's concern. We would like to further discuss the post-PAM interactions. First, the reason why the crystal structure data do not show the post-PAM interaction is that short DNA was used (DOI: 10.1016/j.cell.2015.08.007). Only 2 bp of dsDNA after the PAM were provided, but the detected interaction site in our study is located ~ 6 bp downstream of the PAM. Second, it is unlikely that the post-PAM interaction is due to the flexibility and movement of the protein. If this is the case, this interaction site should also be detected on the opposite site of the PAM or even disappear in some traces. However, this is opposite to what we observed. Third, in addition to SaCas9, we have also detected a similar post-PAM interaction for SpCas9 (DOI: 10.1126/sciadv.aaw9807) and Cpf1 (data not shown). We believe that this is a conserved feature of these proteins. Fourth, a crystal structure study on SpCas9 has indeed demonstrated the existence of its post-PAM interaction with DNA using a long DNA template (DOI: 10.1038/s41467-017-01496-2), which is consistent with the results of our previous study. Fifth, we now have preliminary data proving that this interaction is due to the bending of DNA, and this study is still ongoing (data not shown). Overall, we are confident in concluding that the post-PAM interaction between SaCas9 and DNA indeed exists and functions.

6. On page 12 the authors talk about "traces with only four matches showing the absence of stable binding." However they do not show such traces. Similarly, they talk about traces showing stable binding, but such traces are never shown.

Response: We apologize for not showing these representative traces and have incorporated them into the revised manuscript as the new Fig EV4A.

7. In Figure S7 the authors show the percentage of molecules bound vs the mismatch extent. It seems that an abrupt major transition happens between 4-6 nt, however data for the middle point (5nt) at which the transition may be happening and give more insight into the behaviour of SaCas9 are missing.

Response: We greatly appreciate the suggestion from the referee and have performed the suggested experiment with 5-nt matched sgRNA. It turns out that with 5 bp PAM-proximal matches, dSaCas9 is not able to stably bind to the DNA. These data have been incorporated into the new Fig 3A and Fig EV4A.

8. In Figure 3C the authors should show the full FRET trace including the donor and acceptor signals, as well as a reference trace showing the FRET behaviour of naked DNA.

Response: We thank the referee for the suggestion and have decided to remove all FRET data. We refer the referee to our response to the first comment from referee #3 for more details.

9. There is no reporter dye on any part of the SaCas9:sgRNA complex which would make it possible to observe binding. Consider placing a dye on the sgRNA instead of the target strand to monitor the arrival of Cas9 as well as unzipping behaviour.

Response: We would like to explain the motivation behind our previous FRET assay. The aim of the FRET experiments was to detect the unwinding of the protospacer DNA. Placing a dye on the sgRNA can definitely provide information on the arrival of SaCas9. However, it is not guaranteed that the unwinding can be simultaneously monitored. For example, a FRET study on SpCas9 with labeled sgRNA did not show the unwinding signal (DOI:10.1038/ncomms12778). We thank the referee for the suggestion. Instead of the FRET study, we used a more sensitive probe to report on DNA unwinding and employed a stopped-flow assay for measurement (New Fig 3C and D). For more details, we refer the referee to our response to the first comment from referee #3.

10. Several other smFRET papers on SpCas9 binding have shown fluctuations upon binding and unzipping of the DNA duplex (doi:10.1038/ncomms13350), <https://doi.org/10.1021/jacs.8b03102> . Since the core folds of both SaCas9 and SpCas9 are quite similar, it is surprising that the authors did not observe any dynamic behaviour, either for partially mismatched or full targets, even though the FRET histograms in figure 3D for partially mismatched targets are ambiguous and can contain more than 2 FRET peaks. Please comment.

Response: A minority of our FRET traces with mismatched sgRNAs indeed showed fluctuations, but those with fully matched sgRNA did not. The reasons why we previously chose not to show these data are as follows. 1) The dynamics of DNA unwinding by SaCas9 is not a focus of this work. We simply wanted to examine the DNA unwinding dependence on the number of mismatches between sgRNA and DNA target. 2) The fluctuating traces are not the majority. Even with mismatched sgRNA, only less 15% of the traces showed the fluctuations. We speculate that this effect

might have been due to the labeling position of the dyes on the DNA. Again, we have now decided to replace the FRET data with results from a stopped-flow assay.

Minor comments

1. The authors use somewhat strange English expressions. The text is somehow understandable, but it does not read smoothly at all. It would be greatly beneficial if the manuscript can be revised by a native English speaker with expertise in biology. Correct typos and consider heavily copy-editing the text as some words are missing and some sentences are difficult to follow.

Response: We apologize for the inconvenience and have consulted an English language editor to further improve the language after revising the manuscript. We hope that the referee is satisfied with this revised manuscript.

2. The language used in the manuscript is confusing: the use of pre-PAM and post-PAM interactions indicate interactions that happen prior to PAM binding and following PAM recognition as pre- and post- usually refer to time, rather than position. Consider replacing them with upstream/downstream.

Response: We thank the referee for pointing out that the terminology that we used might not be accurate. We would like to explain the choice of this terminology. In the forward DNA unzipping assay (Fig 1B), we unzipped the DNA from the upstream side of the PAM. The pre-PAM interaction was detected before we unzipped the PAM in terms of time. Consequently, it was termed a pre-PAM interaction, and the other interaction was accordingly termed a post-PAM interaction. In a previous work by our lab (DOI: 10.1126/sciadv.aaw9807), similar terminology was used. To be consistent, we would like to keep it that way. We truly hope that the referee can understand that.

3. The authors' claim that they show results with "near base-pair resolution" is too strong for the data they present as an uncertainty of 4-10 bp in some of their measurements is 25%-50% of the total DNA length that is of interest (length of the target).

Response: We apologize for the inaccuracy. We toned down the statement and used "a few-base-pair resolution" instead.

4. In Introduction it is mentioned that sgRNA comprises crRNA and tracrRNA which is factually incorrect as sgRNA does not contain crRNA and tracrRNA but is a chimera of the two.

Response: We apologize for the inaccuracy and have rephrased this portion.

5. Fig. 3A: please indicate how many molecules were tested to calculate the percentages.

Response: These information has been provided in the figure legends, and source data have also been provided.

6. In figure 3B the error bars are so large that essentially there is no real difference between any of the constructs.

Response: The disruption of DNA-bound protein by unzipping DNA is a thermally activated off-equilibrium process. Therefore, the disruption force varied from trace to trace. It is impossible to lower the error bars by collecting more data. We agree with the referee that no significant difference between any of the constructs exists. In fact, we included a statement indicating that neither the post-PAM interaction nor the pre-PAM interaction shows obvious dependence on the RNA-DNA mismatches.

7. Fig. 3D: 3D presentation of multiple graphs looks fancy but can be somewhat deceiving because it is very difficult to check the shape of the histogram. Also a quantitative comparison between the peaks from the graph is not possible. It is always a better practice to present a data set in a way that the values can be directly readable from the graph.

Response: We thank the referee for the suggestion and have replaced the FRET data.

8. Consider adding a supplementary figure that would show the Gaussian fits of the histograms in Figure 3D for clarity.

Response: We thank the referee for the suggestion and have decided to remove all FRET data.

9. Fig. 4C right: Why is there only one bead present with WT SaCas9? Wasn't it a dual trap optical tweezers?

Response: We apologize for not being explicit in describing this experiment. In this assay, we aimed to examine the products of DNA cleavage by SaCas9. If a DNA molecule is cleaved in the middle and partially released by the endonuclease, it will not be attached to two beads via its two ends. Therefore, instead of using two traps, we used a single trap in this assay. The cleaved DNA was extended by laminar flow instead. In this case, both the cleaved DNA products and DNA-bound SaCas9 can be monitored via fluorescent signals. We have revised our manuscript to clarify this point.

10. Fig. S12: I can only assume that the data in Fig. S12 is measured without SaCas9.

Is it correct? Also, in the legend, they mentioned about Cy5 signal. Wasn't it Cy3 they labeled the crRNA? Also, there is a typo (crDNA>crRNA)

Response: It is correct that the data in old Fig S12 were taken in the absence of SaCas9. We have provided this information in the legend. The 5' end of crRNA is indeed labeled with Cy3. We apologize for these typos and have corrected them in the revised manuscript.

11. Fig. S2: Please specify how many DNA molecules they measured.

Response: These data have been provided in the figure legends, and source data have also been provided.

12. On page 3 the authors mention that SaCas9 and SpCas9 share 17% sequence similarity, but do not mention that despite the low sequence similarity the two proteins share very similar core folds.

Response: We have incorporated the suggested statement into the revised manuscript.

13. On page 3, it is written that "ensuing the complementarity of the 20 bp RNA-DNA hybrid, SpCas9 undergoes a significant structural rearrangement to adopt a bilobed architecture". This is incorrect, SpCas9 adopts a bilobed architecture after binding sgRNA. The global structural rearrangements after binding the target DNA are minimal (not counting the domain movements required for cleavage).

Response: We thank the referee for pointing out the inaccurate statement. We have corrected the error in the revised manuscript.

Referee #3:

In this work, Zhang & Zhang et al. probe energetic features of the RNA-guided interactions of SaCas9 with a DNA target using a series of optical tweezers experiments, complemented by biochemical assays that provide information about enzyme catalytic efficiency and FRET assays claimed to provide information about DNA unwinding. Having already applied many of these techniques to SpCas9 in (Zhang et al., Science Advances, 2019), this group has implemented the same experimental workflow on SaCas9 in the present study, allowing comparison of the two Cas9 orthologs. In the context of growing interest in the biotechnological use of SaCas9, the data presented in this work provide a catalog of biophysical parameters for SaCas9 that were previously only known for SpCas9. The authors demonstrate a few differences between the two orthologs that may have

biotechnological significance, which include:

- Catalytically inactive SaCas9 (dSaCas9) cannot be dissociated from DNA targets by processive molecular motors that were able to dissociate dSpCas9 under equivalent conditions. This result suggests that dSaCas9 may remain bound to DNA longer than dSpCas9 in vivo.
- WT SaCas9 releases the PAM-distal DNA fragment following cleavage, whereas SpCas9 remains bound to both product fragments. Given the recently reported multiple-turnover DNA cleavage activity of SaCas9 (compare SpCas9, which conducts single-turnover DNA cleavage), this result suggests that substrate cycling may begin with fast dissociation of the PAM-distal fragment.

Response: We thank the referee for his/her positive remarks and have addressed the individual comments below.

Major concerns:

1. The authors use a FRET assay similar to that described in Singh et al. (2018) NSMB to make the claim that SaCas9 requires 14 PAM-proximal matches between the crRNA and the DNA target to achieve DNA "unwinding." There are problems with the authors' experimental design and their interpretation of the data. I will first explain the informative experiments performed by Singh et al., and then I will explain why the analogous experiments performed by Zhang & Zhang et al. are not informative.

Singh et al. performed their experiments using constructs with both dyes conjugated within the target sequence-this experimental setup was designed to detect the presence of an intermediate unwinding state hypothesized to have somewhere between 8 and 16 PAM-proximal DNA base pairs unwound. Singh et al. then measured differences in the relative population of the intermediate/fully-unwound states on high-fidelity SpCas9 mutants bound to DNA targets with up to 4 PAM-distal mismatches. They found that, in the presence of PAM-distal mismatches, the intermediate unwinding state was more highly populated on the high-fidelity mutants than on the wild-type protein, suggesting a conformational basis for slowed cleavage of mismatched DNA targets by high-fidelity Cas9 mutants.

Zhang & Zhang created a DNA construct that contained a Cy3-dT at position 13 of the target strand and a Cy5-dT at position 24. This construct would not be expected to yield a change in FRET unless unwinding proceeded past position 13 of the target sequence. Therefore, without even looking at the data, it is confusing why the authors conducted the experiments with the 5-22mm, 7-22mm, 9-22mm, and 11-22mm constructs (which do not have enough RNA complementarity to allow DNA unwinding past position 13) in the first place.

Additionally, it is difficult to explain why the 7-22mm, 9-22mm, and 11-22mm experiments do in fact contain significant low-FRET populations. It is possible that in these populations, the dyes are pressed against the protein and consequently

are restricted in their spatial and/or orientational freedom to a low-FRET ensemble.

Therefore, the only experiment with an interpretable result is the comparison of the "Full match" construct to the "15-22mm" construct.

With this comparison alone, it is difficult to ascertain whether the FRET distribution in the 15-22mm experiment represents a meaningful partition between two stable unwound states or a product of the same assay artifact that yielded the results for the constructs containing more mismatches. Furthermore, even in a best-case scenario where the authors were able to definitively identify a stable intermediate unwinding state, the scientific value of this finding would rely on extensive additional experimentation linking the result to propensity for off-target cleavage (which was the main point of Singh et al. 2018), which is not relevant to the remainder of the present manuscript. Why did the authors choose to monitor this particular reaction coordinate, and what does it tell us about the mechanism of SaCas9? In the absence of a strong answer to these questions, I would advise omitting the FRET data and corresponding conclusions from the manuscript.

Response: We greatly appreciate the referee's explanation and suggestion. We will first explain the motivation behind our previous FRET experiments. This is followed by the details of how we revised the manuscript regarding DNA unwinding by SaCas9.

In this work, we aim to provide a complete dynamic picture of SaCas9 in its interplay with DNA. This process can be divided into DNA target binding, unwinding (R-loop formation), cleavage, and dissociation. The motivation behind the FRET experiments was to provide insight into the mechanism of DNA unwinding by SaCas9. As the referee pointed out, a nearly complete unwinding of the protospacer works as a checkpoint for Cas9 to cleave the DNA. We thus asked a question of how many RNA-DNA matches it takes to FULLY unwind the protospacer DNA. This will allow us to correlate the DNA unwinding to cleavage. We would like to emphasize that unlike the NSMB work, we were not trying to identify the unwinding intermediates in this case.

It is well known that DNA unwinding by Cas9 proteins proceeds directionally from the PAM-proximal region to the PAM-distal region. Thus, in our previous manuscript, we chose to place the fluorescent dyes near the PAM-distal region. The detected FRET signal would possibly imply the unwinding of both the PAM-distal DNA and the PAM-proximal DNA. After reading the comment from the referee, we feel that it is possible that the detected FRET signal may come from the spatial rotation of the dye due to the binding of the protein instead of DNA unwinding.

To precisely detect the dependence of DNA unwinding by SaCas9 on RNA-DNA matches, we instead employed a stopped-flow assay with a DNA substrate that

included 2-AP at position 18 of the non-target strand. The fluorescence of 2-AP is strongly quenched by stacking when it is base-paired, making 2-AP a sensitive probe for monitoring base-pairing changes in Cas9. It has already been used to detect the R-loop formation of SpCas9 and Cpf1 (DOI: 10.1016/j.molcel.2018.06.043; DOI: 10.1016/j.celrep.2017.12.041). This method is more accurate in detecting DNA unwinding compared to FRET, as base-pairing changes are reflected by a single sensitive probe. Moreover, it also allows us to measure the rate of R-loop formation. We found that fully matched sgRNA and 19-22 mm sgRNAs can efficiently unwind the protospacer. However, the 2-AP fluorescence only slightly increased in the presence of 6 or more mismatched sgRNA, followed by a slow decrease, indicating an unstable R-loop formation. These data indicate that up to 18 bp matches are required for SaCas9 to efficiently unwind DNA. In addition, unwinding (R-loop formation) rates were also provided for fully matched and partially matched sgRNAs.

This set of data has now been added as the new Fig 4C and D, and the FRET data have been removed, as suggested.

2. The remainder of the data support the main conclusions of the paper, and I believe they can stand alone in the absence of the FRET data. However, the main text of the paper is rife with smaller conclusions or explanations that are presented with imprecise language and/or an unclear experimental basis, due in some cases to poor writing quality and in others to overinterpretation (some examples are presented in "minor concerns"). These problems are so widespread that I consider the text, collectively, to be a major concern. Extensive care should be taken to improve the clarity and precision of the writing throughout the manuscript.

Response: We apologize for the inaccuracy and overstatements in our previous manuscript. We have carefully revised the manuscript by precisely describing the experiments and results and rigorously drawing the conclusions. We have submitted this manuscript for editing by an English language editor to improve the language. With these efforts, we are confident that a qualified manuscript is provided this time. We thank the referee for pointing out the minor concerns. We have addressed them in the following text.

Minor concerns:

1. The interpretations of the unzipping experiments, which are the cornerstone of this paper, require more careful consideration of the physical nature of the events that occur when the unzipping fork reaches the Cas9 complex. The authors invoke a conceptual framework described in Koch et al., Biophys J, 2002, in which proteins bound to double-helical DNA substrates undergo cooperative dissociation events in response to force-induced DNA base-pair disruption. The described experimental setup, when applied to Cas9, requires a completely different mode of interpretation due to Cas9's R-loop-mediated interaction with DNA. Within the complex, the DNA is not even base paired, eliminating base-pair disruption as a

relaxation pathway. Instead, given the relatively loose association of the non-target strand with the protein in known Cas9 structures, the system probably relaxes (in response to applied force) through breakage of interactions between the non-target strand and the protein (within the R-loop) and/or the target strand (within the PAM). It may even be the case that the non-target strand is torn out of the complex without disrupting Cas9's crRNA-mediated association with the target strand, as it is known that SaCas9 can stably bind to a single-stranded target strand (this possibility invalidates the statement "...indicating that the ternary complex was completely disrupted"). I encourage the authors to consult "Direct Observation of the Formation of CRISPR-Cas12a R-loop Complex at the Single-Molecule Level," Cui et al., ChemComm, 2020, for an explanation of the likely relaxation pathway for a similar system. The authors should adjust their writing throughout the manuscript to explain that their data cannot be interpreted in the same way as data for proteins that bind double-helical DNA.

Response: We greatly appreciate the referee's comment on the interpretations of our DNA unzipping experiments. The essence of this DNA unzipping technique is the separation of two ssDNA molecules. These two molecules can be connected by hydrogen bonds or a protein. For naked DNA, the unzipping fork breaks the hydrogen bonds between two ssDNA molecules and results in a characteristic force signature, which is determined by the underlying DNA sequence. In the presence of bound proteins, this gently varying baseline is interrupted by sharp force rises. Breakage is supposed to occur at the weakest interaction site at the fork. For proteins that bind to double-helical DNA, the separation of the DNA-bound dsDNA is further impeded, as the protein clamps onto the DNA, and the unzipping fork actually breaks the protein-dsDNA interaction. For SaCas9, as the protospacer DNA has already been separated before the unzipping, the unzipping fork indeed breaks either the Cas9-ssDNA (non-target strand) or Cas9-DNA/RNA hybrid (target) interaction. Currently, we are not able to differentiate between the two. However, we do not think that the pre-PAM interaction occurs within the PAM because this has never been detected in reverse unzipping experiments with WT SaCas9 (Fig 4B). We also agree with the referee that there is a possibility that SaCas9 may still associate with one of the DNA strands after the disruption. We were very careful when interpreting our data in the revised manuscript.

2. The interpretation problem described in Comment #1 also poses a problem for the presentation of data in which "position" is expressed in terms of base pairs (ex. Fig. 1B). It is physically meaningless to describe a force peak's "position" within the R-loop in terms of base pairs, as base pairs are not being broken. Given the complex and unknown nature of the relaxation pathway, it may even be physically meaningless to ascribe any sort of "position" to a force peak at all (relaxation may involve simultaneous breakage of interactions at multiple locations within the complex). If one assumed that the relaxation pathway involved sequential breakage of non-target-strand:protein interactions along an axis parallel to that of

the RNA:DNA hybrid, one could adjust the model to incorporate the compressed length of the A-form helix in one of the arms leading away from the fork-this would imply that the so-called "pre-PAM" interaction is actually closer to the PAM than in the authors' model (and may even be the protein's interaction with the PAM itself). However, with so little known about the relaxation pathway, the authors could simply acknowledge that they are unable to assign a precise spatial "position" to any force peak in the forward unzipping experiments or to intra-R-loop force peaks in the reverse unzipping experiments (note that this uncertainty does not apply to the first force peak of reverse unzipping experiments, which is approached from an unperturbed DNA duplex). This may involve changing the x-axis units of their unzipping plots to a measure of extension length (or a similar model-independent physical distance that is defined by the displacement of the optical tweezers), accompanied by diagrams that indicate the hypothesized "position" of the interaction only in very general terms (ex. "somewhere within the R-loop" or "near the PAM"). The text should also be adjusted accordingly. For example, I find the following sentences unacceptable because they reveal a failure to correctly interpret what can be known about a given interaction based on the extension length at its associated force peak: "The other pre-PAM interaction between dSaCas9 and DNA is located within the DNA-RNA heteroduplex region." "Even though the structural data showed that the C-terminal region of the REC lobe of SaCas9 interacts with the PAM-distal region of the heteroduplex, ..." (Objection: It is unknown whether breakage of a heteroduplex:protein interaction accounts for relaxation of this force peak, and I would argue that it is in fact more likely that we are observing breakage of a non-target-strand:protein interaction. The authors should remain agnostic.)

Response: We agree with the referee that the word "position (in bp)" in the reversed unzipping assays might not be accurate and have revised the manuscript as follows.

We would like to first detail the process of our data processing. In our experiments, once a single DNA tether was located, we moved the coverslip at a constant speed of 50 nm/s. In this process, the force (in pN) and extension (in nm) of DNA as functions of time were simultaneously recorded. The extension of the DNA (X) (in nm) has contributions from both the anchored dsDNA (X_{ds}) and ssDNA (X_{ss}) under the same force. X_{ds} (in nm) is determined because the force-extension curve of the anchoring segment is fully characterized by the WLC model (DOI: 10.1126/science.1439819). X_{ss} (in nm) is thus obtained and is proportional to the number of nucleotides of ssDNA. Using the extensible FJC model (DOI: 10.1126/science.271.5250.795), X_{ss} (in nm) under a known force is converted to the number of nucleotides of ssDNA (j) (in nt). Previously, we further converted j to the number of base pairs unzipped by dividing it by 2 ($j/2$ in bp) based on the assumption that the disruption of one base pair of the protospacer DNA releases two nucleotides.

As the referee can tell from the data analysis, the number of base pairs unzipped still represents the same information before and after the R-loop, as long as the unzipped dsDNA is in the form of ssDNA. To be more precise, we decided to label the X axis as “estimated position (bp)”. In addition, we clearly stated in both the main text and methods section that the position of the pre-PAM interaction is not accurate but is instead an estimation based on the assumption that the protospacer was still intact dsDNA upon binding by dSaCas9. We were also more cautious when talking about this interaction site, and the manuscript was revised accordingly.

3. In Fig. 1E, the gel should not be cropped so tightly at the bottom. The reader should be able to ascertain that the exonuclease did not proceed more deeply into the complex.

Response: We thank the referee for the suggestion and have provided an updated gel picture in the revised manuscript, as suggested (new Fig 1E).

4. In Fig. 2C/D, BLM is depicted to proceed in two different directions. What is the correct polarity of its processive unwinding activity?

Response: We apologize for the mistake. BLM is a 3'-5' helicase. The arrow should be downward in Figure 2C, and we have corrected this error.

5. In Fig. 3F, it is misleading to present single-timepoint cleaved fractions, which reflect an arbitrarily chosen timepoint, on the same scale as the "Binding" and "Unwinding" parameters, which reflect a meaningful equilibrium (ex. the cleavage curve may have looked identical to the binding curve if the authors had chosen a later timepoint). The "Cleavage" data should be depicted in a separate plot or on its own axis that indicates "Fraction cleaved at X timepoint".

Response: We greatly appreciate the suggestion from the referee. We agree that the cleavage activity of SaCas9 strongly depends on the incubation time. We have provided an updated DNA cleavage gel picture with more time points and have removed the old Figure 3F.

6. When I zoom in on the background of the micrographs in Fig. 4C, the Cy3 puncta in the background are much brighter than those in the background of Fig. S12, even though the two should be equivalent based on my understanding of the methods. The authors should make sure they are collecting and presenting their micrographs equivalently.

Response: The brightness of the Cy3 puncta in the background is determined by a pre-set threshold for the number of photons detected. In the revised manuscript, this pre-set value is consistent for all figures.

7. How did the authors verify the following statement? "In addition, the bound dSaCas9's lifetime was longer than 2 hours."

Response: The typical duration of our DNA unzipping experiments is over 2 hours. The binding efficiency of dSaCas9 on DNA after 2 hours was found to be 100% with fully matched sgRNA in our previous experiments. We have conducted a new series of experiments to illustrate that the lifetime of DNA-bound dSaCas9 was actually over 24 hours (new Fig EV2G).

8. In Fig. S4, how many reverse unzipping events were conducted at each Cas9 concentration? This should be indicated in the figure legend.

Response: We thank the referee for the suggestion. These data have been provided in the legends, and source data have also been provided.

9. The first sentence of the intro contains an incorrect statement. "Cas" stands for "CRISPR-associated" (not "CRISPR-associated system") and refers to the protein-coding genes of CRISPR-Cas systems.

Response: We apologize for the inaccuracy and have corrected it in the revised manuscript.

10. The second sentence of the intro contains an incorrect statement. The entity described by the authors is the type II interference complex, not the type II CRISPR-Cas system.

Response: We thank the referee for pointing out the inaccurate statement and have corrected it in the revised manuscript.

11. The last sentence on page 3 contains an incorrect statement. The bilobed architecture of Cas9 is an intrinsic feature of the protein in all known ligand states-it is not only adopted after DNA-induced rearrangement.

Response: We apologize for the incorrect statement and have corrected it in the revised manuscript.

12. The clause "even though the post-PAM interaction between SaCas9 and DNA becomes stronger after cleavage (Supp. Fig. 11)" is an overinterpretation. The increased interaction strength shown in Supp. Fig. 11 could just as well be due to a direct stabilizing effect of Mg²⁺ (and not the release of the PAM-distal fragment). The authors would need to do a Mg²⁺-free experiment with an already-cleaved DNA substrate to verify their claim.

Response: We thank the referee for pointing out the overinterpretation and suggesting the experiment to examine the other possibility. As suggested, we have conducted DNA unzipping experiments with WT SaCas9 in the presence and absence of Mg^{2+} . Without Mg^{2+} , the unzipping signatures with WT SaCas9 resembled those with dSaCas9 and indeed showed a weaker Post-PAM interaction than that observed with Mg^{2+} (new Fig EV5A). Moreover, after DNA cleavage, the flow-in of a Mg^{2+} -free buffer results in the immediate dissociation of PAM-bound SaCas9 (data not shown). These data indicate that the increased post-PAM interaction is possibly due to a stabilizing effect of Mg^{2+} instead of DNA cleavage. We have revised the manuscript and toned down the corresponding statements.

Non-essential suggestions for improving the study:

1. In Fig. 4C, the Cy3 spot on the end of the DNA is barely visible and not so different from other background Cy3 spots. The authors should consider an additional (or alternative) mode of data presentation/analysis that can assure the reader that there is indeed a stable spot there.

Response: We thank the referee for the suggestion. We have shown the corresponding fluorescence intensity alongside the pictures to further highlight the Cy3 spots (new Fig 4C).

2. While the terms "downstream/upstream of the PAM" are useful when defined in a figure, their arbitrary definition makes them confusing when reading text alone and should be avoided as much as possible, especially in the abstract.

Response: We are grateful for this suggestion. We have revised the text to avoid confusion.

3. The authors have questionable word choice in several places (examples: ensuing, testify, intense, formidable, asks for).

Response: We apologize again for the language. The manuscript has been edited by both us and an English language editor several times. We hope that the referee will be satisfied with the revised manuscript.

4. The authors should include a written description of the physical process occurring during the slow linear drop in force following each "dissociation" event. After reading the sentence: "after the rise in force, they continued to be similar to those of the corresponding naked DNA," I was initially confused as to why the protein-containing traces did not resemble the gray traces.

Response: We apologize for not clearly describing our data. Regarding the data presentation and description of our DNA unzipping experiment, we refer the referee to our response to the second question from the referee #2. Briefly, we have shown

complete unzipping traces in the new Fig 1B and C and explained the data in detail in the text.

5. The text in the section titled "dSaCas9 presents a strong barrier to DNA tracking motors" could be significantly condensed because the experimental setup was the same for all motors, and the results were nearly equivalent for all motors.

Response: We appreciate the referee's advice and have shortened this section in the revised manuscript.

-----The end-----

Dear Prof. Sun

Thank you for the submission of your revised manuscript. We have now received the comments from referee 1, who assessed your entire response to all referees, and I am happy to say that we can in principle accept your manuscript now.

Only a few more minor changes are required:

- Your manuscript has 5 main figures and thus classifies as a short report with combined results and discussion sections and a maximum of 20.000 characters (excluding references and materials and methods and including figure legends). If you prefer instead to publish your study as a full article with separate results and discussion sections, you would need to add one main figure to the main manuscript file. Either option is fine.
- Please add up to 5 keywords to the manuscript file
- Please add the author contribution section
- Please change the reference format to Harvard style. You can find a direct link in our guide to authors.
- Please add the funder information into our online system when you upload the new manuscript files
- Please callout either all panels of figure EV3, or the entire figure as a whole.
- Please reduce the contrast on the blots in figures 3 and EV4 to avoid over-contrasting.
- I attach to this email a related manuscript file with comments by our data editors. Please address all comments in the final manuscript file.
- On the synopsis image, at the final size of 550 pixels x 300 pixels, the smallest text is too small. Please increase the font size.
- Please complete the author checklist section B on statistics, as you have a few figures that show statistics.

Please upload with your final manuscript a short summary of your findings and their significance (1-2 sentences) and 3-4 bullet points highlighting key results. This information will be uploaded together with your synopsis image on our EMBO reports homepage.

I am looking forward to seeing a final version of your manuscript as soon as possible. Please let me know if you have any questions.

Esther Schnapp, PhD

Senior Editor
EMBO reports

Referee #1:

The authors have addressed the comments from the reviewers in great detail. The revisions and additional experiments including a stopped-flow assay with a DNA substrate containing 2-aminopurine have greatly improved the manuscript. The revised manuscript is a good fit for EMBO reports and will contribute new knowledge to the field.

The authors have completed all minor editorial requests.

Prof. Bo Sun
ShanghaiTech University
School of Life Science and Technology
393, Central Huaxia Road
Pudong Distinct
Shanghai, Shanghai 201210
China

Dear Prof. Sun,

I am very pleased to accept your manuscript for publication in the next available issue of EMBO reports. Thank you for your contribution to our journal.

At the end of this email I include important information about how to proceed. Please ensure that you take the time to read the information and complete and return the necessary forms to allow us to publish your manuscript as quickly as possible.

As part of the EMBO publication's Transparent Editorial Process, EMBO reports publishes online a Review Process File to accompany accepted manuscripts. As you are aware, this File will be published in conjunction with your paper and will include the referee reports, your point-by-point response and all pertinent correspondence relating to the manuscript.

If you do NOT want this File to be published, please inform the editorial office within 2 days, if you have not done so already, otherwise the File will be published by default [contact: emboreports@embo.org]. If you do opt out, the Review Process File link will point to the following statement: "No Review Process File is available with this article, as the authors have chosen not to make the review process public in this case."

Should you be planning a Press Release on your article, please get in contact with emboreports@wiley.com as early as possible, in order to coordinate publication and release dates.

Thank you again for your contribution to EMBO reports and congratulations on a successful publication. Please consider us again in the future for your most exciting work.

Yours sincerely,

THINGS TO DO NOW:

You will receive proofs by e-mail approximately 2-3 weeks after all relevant files have been sent to

our Production Office; you should return your corrections within 2 days of receiving the proofs.

Please inform us if there is likely to be any difficulty in reaching you at the above address at that time. Failure to meet our deadlines may result in a delay of publication, or publication without your corrections.

All further communications concerning your paper should quote reference number EMBOR-2020-50184V3 and be addressed to emboreports@wiley.com.

Should you be planning a Press Release on your article, please get in contact with emboreports@wiley.com as early as possible, in order to coordinate publication and release dates.

Corresponding Author Name: Bo Sun

Manuscript Number: EMBOR-2020-50184